# Neural stem cells traffic functional mitochondria via extracellular vesicles

**Luca Peruzzotti-Jametti**[1]⊗*, **Joshua D. Bernstock**[1,2]⊗, **Cory M. Willis**[1],
**Giulia Manferrari**[1], **Rebecca Rogall**[1], **Erika Fernandez-Vizarra**[3], **James
C. Williamson**[4,5], **Alice Braga**[1], **Aletta van den Bosch**[1], **Tommaso Leonardi**[1,6],
**Grzegorz Krzak**[1], **Ágnes Kittel**[7], **Cristiane Benincá**[3], **Nunzio Vicario**[1,8],
**Sisareuth Tan**[9], **Carlos Bastos**[10], **Iacopo Bicci**[1], **Nunzio Iraci**[1,8], **Jayden A. Smith**[11],
**Ben Peacock**[12], **Karin H. Muller**[13], **Paul J. Lehner**[4,5], **Edit Iren Buzas**[14,15,16],
**Nuno Faria**[10], **Massimo Zeviani**[3], **Christian Frezza**[17], **Alain Brisson**[9], **Nicholas
J. Matheson**[4,5,18], **Carlo Viscomi**[3], **Stefano Pluchino**[1,11]*

1 Department of Clinical Neurosciences and NIHR Biomedical Research Centre, University of Cambridge,
United Kingdom, 2 National Institutes of Health (NINDS/NIH), Bethesda, Maryland, United States of America,
3 MRC Mitochondrial Biology Unit, University of Cambridge, United Kingdom, 4 Cambridge Institute of
Therapeutic Immunology and Infectious Disease (CITIID), University of Cambridge, Cambridge, United
Kingdom, 5 NHS Blood and Transplant, Cambridge, United Kingdom, 6 Center for Genomic Science of
IIT@SEMM, Istituto Italiano di Tecnologia (IIT), Milan, Italy, 7 Institute of Experimental Medicine, Eötvös
Lorand Research Network, Budapest, Hungary, 8 Department of Biomedical and Biotechnological Sciences
(BIOMETEC), University of Catania, Italy, 9 UMR-CBMN CNRS-Université de Bordeaux-IPB, France,
10 Department of Veterinary Medicine, University of Cambridge, Cambridge, United Kingdom, 11 Cambridge
Innovation Technologies Consulting (CITC) Limited, United Kingdom, 12 NanoFCM Co., Ltd, Nottingham,
United Kingdom, 13 Cambridge Advanced Imaging Centre (CAIC), United Kingdom, 14 Semmelweis
University, Budapest, Hungary, 15 HCEMM Kft HU, Budapest, Hungary, 16 ELKH-SE, Budapest, Hungary,
17 MRC Cancer Unit, Hutchison/MRC Research Centre, University of Cambridge, Cambridge United
Kingdom, 18 Department of Medicine, University of Cambridge, United Kingdom

⊗ These authors contributed equally to this work.
* lp429@cam.ac.uk (LP-J); spp24@cam.ac.uk (SP)

pbio.3001166

Dundee, UNITED KINGDOM

**Data Availability Statement:** The microarray raw
data were deposited in ArrayExpress with the
accession number E-MTAB-8250. The proteomic
data described in this study have been deposited to

## Abstract

Neural stem cell (NSC) transplantation induces recovery in animal models of central ner-
vous system (CNS) diseases. Although the replacement of lost endogenous cells was origi-
nally proposed as the primary healing mechanism of NSC grafts, it is now clear that
transplanted NSCs operate via multiple mechanisms, including the horizontal exchange of
therapeutic cargoes to host cells via extracellular vesicles (EVs). EVs are membrane parti-
cles trafficking nucleic acids, proteins, metabolites and metabolic enzymes, lipids, and entire
organelles. However, the function and the contribution of these cargoes to the broad thera-
peutic effects of NSCs are yet to be fully understood. Mitochondrial dysfunction is an estab-
lished feature of several inflammatory and degenerative CNS disorders, most of which are
potentially treatable with exogenous stem cell therapeutics. Herein, we investigated the
hypothesis that NSCs release and traffic functional mitochondria via EVs to restore mito-
chondrial function in target cells. Untargeted proteomics revealed a significant enrichment
of mitochondrial proteins spontaneously released by NSCs in EVs. Morphological and func-
tional analyses confirmed the presence of ultrastructurally intact mitochondria within EVs
with conserved membrane potential and respiration. We found that the transfer of these
mitochondria from EVs to mtDNA-deficient L929 Rho[0] cells rescued mitochondrial function

the ProteomeXchange consortium via the PRIDE partner repository (accessible at http://proteomecentral.proteomexchange.org) with the dataset identifier PXD024368. Remaining relevant data are within the paper and its Supporting Information files.

**Funding:** This work was funded by the Italian Multiple Sclerosis Association (AISM, grant 2010/R/31 and grant 2014/PMS/4 to SP), the Italian Ministry of Health (GR08-7 to SP), the European Research Council (ERC) under the ERC-2010-StG grant agreement n° 260511-SEM_SEM, the Medical Research Council (CSF MR/P008801/1 to NJM), NHS Blood and Transplant (WPA15-02 to NJM), the Engineering and Physical Sciences Research Council, the Biotechnology and Biological Sciences Research Council UK Regenerative Medicine Platform Hub "Acellular Approaches for Therapeutic Delivery" (MR/K026682/1 to SP), the Evelyn Trust (RG 69865 to SP), the Bascule Charitable Trust (RG 75149 to SP), the Wellcome Trust (RRZA/057 RG79423 to LPJ and PRF 101835/Z/13/Z to PJL), the Addenbrooke's Charitable Trust (RG 97519 to LPJ), the NIHR Cambridge BRC, and by a Wellcome Trust Strategic Award to CIMR. LPJ was supported by a senior research fellowship from FISM - Fondazione Italiana Sclerosi Multipla (cod. 2017/B/5) financed or co financed with the '5 per mille' public funding, and by a Wellcome Trust Clinical Research Career Development Fellowship (RRAG/214). RR was funded by the german Fritz Thyssen Foundation with a one-year fellowship (40.16.0.026MN). EFV, CB and MZ are funded by a core grant of the MRC to the Mitochondrial Biology Unit (MC_UU_00015/5), ERC Advanced Grant (FP7-322424 to MZ) and NRJ-Institut de France (to MZ). The funders had no role in study design, data collection and analysis, decision to publish, or preparation of the manuscript.

**Competing interests:** I have read the journal's policy and the authors of this manuscript have the following competing interests: SP is co-founder, CSO and shareholder (>5%) of CITC Ltd. and iSTEM Therapeutics Litd., and co-founder and Non-executive Director at asitia Therapeutics Ltd.; LPJ is shareholder of CITC Ltd.; JAS is a Project Manager and Senior Research Associate at CITC Ltd. and Director of Research of iSTEM Therapeutics Ltd.; BP is an employee of NanoFCM and his contributions to this paper were made as part of their employment.

**Abbreviations:** ACN, acetonitrile; AWERB, Animal Welfare and Ethical Review Body; bFGF, basic fibroblast growth factor; BN-PAGE, blue native polyacrylamide gel electrophoresis; BSA, bovine

and increased Rho$^0$ cell survival. Furthermore, the incorporation of mitochondria from EVs into inflammatory mononuclear phagocytes restored normal mitochondrial dynamics and cellular metabolism and reduced the expression of pro-inflammatory markers in target cells. When transplanted in an animal model of multiple sclerosis, exogenous NSCs actively transferred mitochondria to mononuclear phagocytes and induced a significant amelioration of clinical deficits. Our data provide the first evidence that NSCs deliver functional mitochondria to target cells via EVs, paving the way for the development of novel (a)cellular approaches aimed at restoring mitochondrial dysfunction not only in multiple sclerosis, but also in degenerative neurological diseases.

## Introduction

Extracellular vesicles (EVs) are a heterogeneous population of secreted membrane vesicles with distinct biogenesis, biophysical properties, and functions, which are common to virtually all cells and life forms [1]. Despite their proven biological potential, the characterisation and classification of this heterogeneous population of membrane vesicles has thus far been challenging. A working basis for a classification system of EVs is to divide them into 3 major subtypes based on biogenic, morphological, and biochemical properties: exosomes, microvesicles (MVs), and apoptotic bodies [2]. Exosomes are small vesicles, ranging 30 to 150 nm in diameter, generated from the inward budding of intracellular multivesicular bodies and released after the subsequent fusion with the plasma membrane [3]. MVs are membranous vesicles generated by clathrin-mediated shedding of the plasma membrane and released into the extracellular space, with a diameter ranging 50 to 1,000 nm [3]. Apoptotic bodies are generated through apoptotic fragmentation and blebbing with a resultant size range of 1,000 to 5,000 nm [4]. As the classification of EVs is continuously evolving, recent consensus has further simplified nomenclature by dichotomising EVs into 2 major categories: ectosomes, for particles released through plasma membrane budding; and exosomes, for particles originated from the endosomal pathway [5].

Neural stem cells (NSCs) are classically defined as a heterogeneous population of self-renewing, multipotent stem cells of the developing and adult central nervous system (CNS), which reside within specialised microenvironments and drive neurogenesis and gliogenesis [6,7]. Data from our lab and peers have shown that in addition to the (expected) cell replacement, NSCs are strikingly able to engage in multiple mechanisms of action in the diseased CNS [6,8], including the horizontal exchange of therapeutic cargoes to host cells via EVs [9]. However, the function and contribution of these cargoes to the broad therapeutic effects of NSCs are not fully understood.

We have recently focused on defining the nature and function of intercellular signalling mediated by EVs from NSCs [10,11]. Using a series of computational analyses and high-resolution imaging techniques, we have demonstrated that EVs deliver functional interferon gamma/interferon gamma receptor 1 (IFNγ/Ifngr1) complexes to target cells [10]. We also discovered that EVs are endowed with intrinsic metabolic activities and harbour selective L-asparaginase activity catalysed by the enzyme asparaginase-like protein 1 [11].

Recent evidence suggests that mitochondria play a key role in intercellular communications and that the release of mitochondria (or mitochondrial components) into the extracellular space has important functional consequences [12,13]. Growing attention has been given to the mechanisms regulating the mitochondrial exchange between cells. Several processes have been described, including the formation of actin-based tunnelling nanotubes [14], cell-to-cell contact via gap junctions [15], and the release of EVs [16]. The latter mechanism seems to be

serum albumin; CBS, concentric backscatter detector; CCCP, carbonyl cyanide m-chlorophenyl hydrazone; cDNA, complementary DNA; CGM, complete growth media; CLEM, correlative-light electron microscopy; CNS, central nervous system; cryo-TEM, cryo-transmission electron microscopy; CT, threshold cycle; D/P, Dynasore and Pitstop 2; DAB, diaminobenzidine tetrahydrochloride; DAPI, 4′,6-diamidino-2-phenylindole; DDM, n-dodecyl-β-D-maltoside; DHODH, dihydroorotate dehydrogenase; DIW, deionised water; dpi, days post immunisation; EAE, experimental autoimmune encephalomyelitis; ECAR, extracellular acidification rate; EGF, epidermal growth factor; ETC, electron transport chain; EV, extracellular vesicle; FACS, fluorescent-activated cell sorting; FBS, fetal bovine serum; FCCP, carbonyl cyanide-4-(trifluoromethoxy)phenylhydrazone; FDR, false discovery rate; fGFP, farnesylated green fluorescent protein; FP, fluorescent protein; GAGE, Generally Applicable Gene-set Enrichment; GFP, green fluorescent protein; GOCC, Gene Ontology Cellular Component; HpRP, high pH reversed phase; HRR, high-resolution respirometry; ICV, intracerebroventricularly; IFNγ, interferon gamma; Ifngr1, interferon gamma receptor 1; i.p., intraperitoneal; KEGG, Kyoto Encyclopedia of Genes and Genomes; LPS, lipopolysaccharide; Mφ, macrophages; MDV, mitochondria-derived vesicle; MOG, myelin oligodendrocyte glycoprotein; MSC, mesenchymal stem cell; mt-ND1, mitochondrial gene NADH dehydrogenase subunit 1; MV, microvesicle; NanoFCM, nano flow cytometry; NBT, nitro blue tetrazolium; NP, nanoparticles; NSC, neural stem cell; NTA, nanoparticle tracking analysis; OCR, oxygen consumption rate; OCT, optimum cutting temperature; OXPHOS, oxidative phosphorylation; PCR, polymerase chain reaction; PD, peak of disease; PFA, paraformaldehyde; PMS, phenazine methasulfate; qPCR, quantitative gene expression analysis; ROI, region of interest; RT, room temperature; Sdhd, succinate dehydrogenase complex subunit D; SDS, sodium dodecyl sulfate; SEM, scanning electron microscopy; SPCM APD, single-photon counting avalanche photodiodes detector; SVZ, subventricular zone; TEM, transmission electron microscopy; TMT, Tandem Mass Tag; TRPS, tunable resistive pulse sensing; WB, western blot; XF, extracellular flux.

central in regulating the exchange of mitochondria to inflammatory cells and could represent a novel mechanism of immunomodulation [17–19].

These pivotal observations prompted us to investigate whether EVs released by NSCs also harbour mitochondria and what their functional relevance is for intercellular communication, immune modulation, and tissue repair.

Here, we used an untargeted Tandem Mass Tag (TMT)-based proteomic analysis to investigate the protein content of EVs that are released by NSCs in vitro. We found an enrichment of mitochondrial proteins in both unfractionated EV preparations and exosome-specific fractions. Morphological and functional analyses unveiled structurally, and functionally intact, free and EV-encapsulated mitochondria. We next studied the transfer of these extracellular mitochondria into target cells and found that they were efficiently incorporated by both somatic cells and mononuclear phagocytes in vitro. Specifically, mitochondria from EVs rescued mitochondrial function in mtDNA-deficient L929 Rho[0] cells, as well as integrated into the host mitochondrial network of inflammatory macrophages (Mφ), thus modifying their metabolic profile and pro-inflammatory gene expression. When NSCs and EVs were intracerebroventricularly (ICV) delivered in mice with myelin oligodendrocyte glycoprotein (MOG)-induced experimental autoimmune encephalomyelitis (EAE), both NSCs and EVs induced a significant amelioration of neurological deficits, and transplanted NSCs transferred mitochondria to mononuclear phagocytes in vivo.

Our data suggest that horizontal transfer of functional mitochondrial via EVs is a mechanism of signalling used by NSCs to modulate the physiology and metabolism of target cells, opening a possible new avenue for the development of acellular therapies aimed at correcting mitochondrial dysfunction in the CNS.

## Results

### Proteomic analysis of EVs and exosomes identifies mitochondrial proteins

We first performed an untargeted multiplex TMT-based proteomic analysis of the whole EV fraction and sucrose gradient–purified exosomes spontaneously released by NSCs in vitro and compared them with parental NSC whole-cell lysates (Fig 1A, S1 Data).

Using Gene Ontology Cellular Component (GOCC) annotations, we investigated the subcellular origin of proteins enriched in EVs compared with NSCs (Fig 1B). We found that proteins with annotations indicating exosomal localisation were markedly enriched in EVs, whereas proteins with annotations indicating nuclear localisation were depleted (Fig 1B). Interestingly, proteins with annotations indicating mitochondrial localisation were also significantly enriched in EVs versus NSCs (Fig 1B).

To further investigate this latter finding, we next examined our data with specific GOCC daughter annotations indicating localisation to the 3 major mitochondrial structural components: outer membrane, inner membrane, and matrix. We found that proteins with these annotations were relatively enriched in EVs versus NSCs (Fig 1C). We also specifically scrutinised the relative abundances of subunits of the 5 mitochondrial complexes and discovered that mitochondrial proteins coded in both the mitochondrial and nuclear genomes were all significantly enriched in EVs (Fig 1D). We then investigated the DNA content of EVs and found that the mitochondrial gene NADH dehydrogenase subunit 1 (*mt-ND1*)—which is encoded in the mtDNA—was present in EVs, regardless of their pretreatment with DNase I. However, this was not the case for the mitochondrial gene succinate dehydrogenase complex subunit D (*Sdhd*), which is encoded in the nuclear DNA (Fig 1E), thus showing that nuclear DNA was not enriched, but instead suggesting the likely presence of mitochondria with intact mitochondrial matrix in EVs.

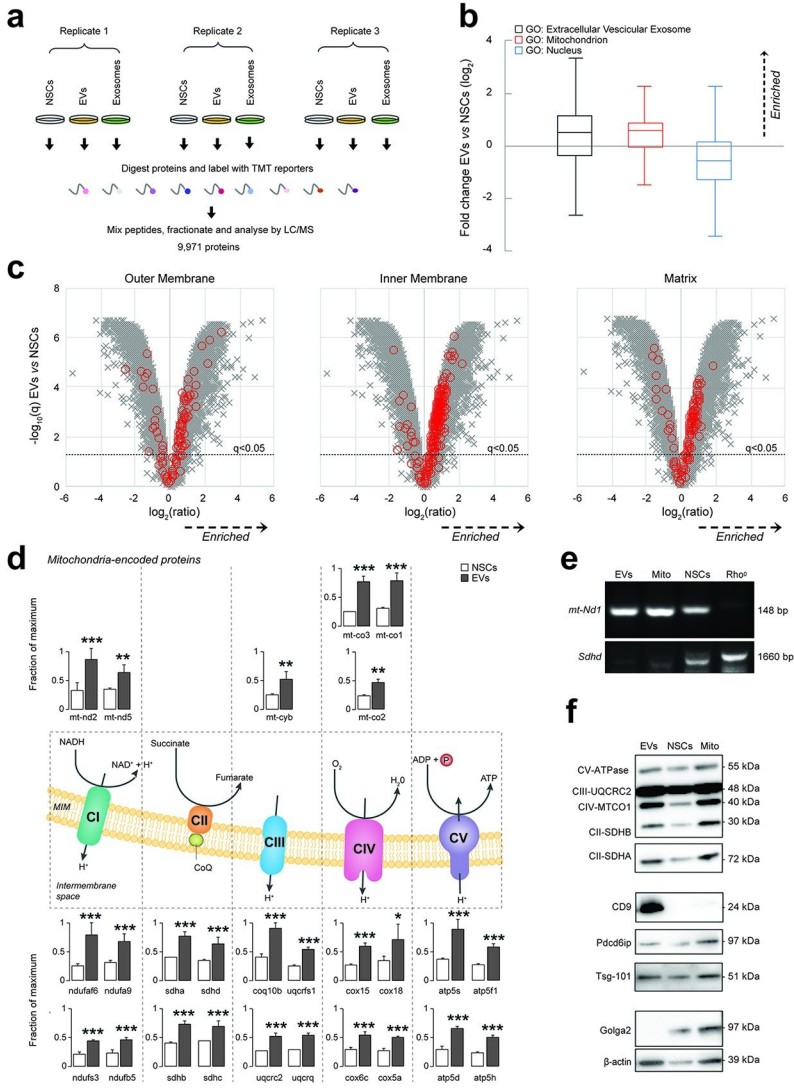

**Fig 1. NSCs shed mitochondrial proteins and mtDNA via EVs. (a)** Overview of multiplex TMT-based proteomic experiment. TMT-based proteomics identified a total of 9,971 proteins, of which 9,951 were quantitated across all conditions. **(b)** Relative abundance of proteins annotated with indicated GOCC subcellular localisations in EVs compared with NSC whole-cell lysates (NSCs). Annotations were available for 9,049/9,971 cellular proteins identified in the multiplex TMT-based functional proteomic experiment illustrated in a. Boxplots show median, interquartile range, and Tukey whiskers for proteins with the following annotations: extracellular vesicular exosome (GO:0070062, black outline, enriched in EVs); mitochondrion (GO:0005739, red outline, enriched in EVs); and nucleus (GO:0005634, blue outline, depleted in EVs). Data are from $N$ = 3 independent biological replicates. (Data available on ProteomeXchange, identifier PXD024368, and in S3 Data). **(c)** Relative abundance of proteins from different mitochondrial compartments (outer membrane, matrix, and inner membrane) in EVs compared with NSCs. Volcano plots show statistical significance (*y* axis) vs. fold change (*x* axis) for 9,951/9,971 cellular proteins quantitated across all 3 biological replicates (no missing values) in the multiplex TMT-based functional proteomic experiment illustrated in a. Proteins annotated with the following GOCC subcellular localisations are highlighted in red: mitochondrial outer membrane (GO:0005741, enriched in EVs); mitochondrial inner membrane (GO: 0005743, enriched in EVs); and mitochondrial matrix (GO:0005759, enriched in EVs). An FDR threshold of 5% is indicated (proteins with Benjamini–Hochberg FDR-adjusted *p*-values (q values) < 0.05). (Data available on ProteomeXchange, identifier PXD024368, and in S3 Data). **(d)** Relative abundance of selected mitochondrial proteins in EVs compared with NSCs. Mitochondrial complex (C) proteins enriched in EVs and encoded in the mitochondrial (upper panel) or nuclear (lower panel) genomes in the multiplex TMT-based functional proteomic experiment include: NADH:ubiquinone oxidoreductase or CI [mtnd2 (ND2 subunit), mtnd5 (ND5 subunit), ndufaf6 (assembly factor 6), ndufa9 (subunit A9), ndufs3 (core subunit S3), ndufb5 (1 beta subcomplex subunit 5)], succinate dehydrogenase or CII [Sdha (Subunit A), sdhd (cytochrome b small subunit), sdhb (iron-sulfur subunit), sdhc (cytochrome b560 subunit)], cytochrome b-c1 or CIII

[mt-cyb (cytochrome B), Uqcrfs1 (subunit 5), Uqcrc2 (subunit 2), coq10b (coenzyme Q10B), uqcrq (subunit 8)], cytochrome C oxidase or CIV [mt-co3 (oxidase III), mtco1 (oxidase I), mtco2 (oxidase II), cox15 (subunit 15), cox18 (assembly protein 18), cox6c (subunit 6C), cox5a (subunit 5a)] and ATP synthase or CV [atp5f1 (subunit gamma), atp5s (subunit S), atp5d (subunit delta), atp5h (subunit D)]. Mean abundances (fraction of maximum) and 95% CIs from $N = 3$ independent biological replicates are shown. *q < 0.05, **q < 0.01, ***q < 0.001 vs. NSCs. (Data available on ProteomeXchange, identifier PXD024368, and in S3 Data). **(e)** Representative PCR amplification of DNA extracted from NSCs, EVs, and isolated mitochondria (Mito). The mitochondrial encoded gene *mt-ND1* (NADH-ubiquinone oxidoreductase chain 1) was found to be present in EVs, Mito, and NSCs (L929 Rho$^0$ were used as negative controls). **(f)** Representative protein expression analysis by WB of NSCs, EVs, and isolated mitochondria (Mito). Mitochondrial complex proteins (CV-ATPase, CII-SDHA, CII-SDHB, CIV-MTCO1, and CIII-UQCRC2), EV positive markers (Tsg-101, Pdcd6ip, and CD9), and negative EV marker (Golga2) are shown, as well as β-actin. CI, II, II, IV, V, complex I, II, II, IV, V; EV, extracellular vesicle; FDR, false discovery rate; GOCC, Gene Ontology Cellular Component; NSC, neural stem cell; PCR, polymerase chain reaction; TMT, Tandem Mass Tag; WB, western blot.

To further validate our TMT-based proteomic data using an orthogonal technique, we next subjected EVs and NSCs to immunoblot analysis. EVs were enriched in exosomal markers (CD9, Pdcd6ip, and Tsg101) and mitochondrial complexes but depleted of Golgi markers (Golga2) (Fig 1F) compared to NSCs. Conversely, a control preparation enriched in mitochondria obtained from NSCs lysates (referred to as Mito) [20] was found to be depleted of CD9 and enriched in Golga2 (Fig 1F).

To exclude any potential bias related to our own purification methods, we further employed 2 additional high-quality and scalable commercially available exosome/EV precipitation-based isolation protocols that avoid ultracentrifugation [21,22] (Fig 2A). We found that these protocols yielded EVs depleted of Golga2 but enriched in CD9 and mitochondrial proteins.

Since it is known that most of these precipitation-based kits also co-isolate non-EV components [23], we then sought to further analyse the NSC-derived EVs using 2 additional protocols. First, we used an immune-mediated isolation kit for the EV marker CD63, thus obtaining a CD63-enriched EV fraction (EVs$^{CD63\_enrich.}$) and a CD63-depleted EV fraction (EVs$^{CD63\_depl.}$) (S1 Fig). We found that the expression of mitochondrial complex proteins was highest in EVs$^{CD63\_enrich.}$ versus EVs$^{CD63\_depl.}$, while both EV and mitochondrial complexes were absent from control media preparations (S1 Fig). Second, since intact mitochondria normally range between 0.2 and 1 μm [24], we tested an ultracentrifugation protocol that adds an additional 0.22 μm ultrafiltration step to the EV isolation [25]. We found that this protocol, used to size-exclude intact mitochondria (hereafter referred to as EVs$^{Mito\_depl.}$), led to a significant reduction in the intensity of signal from mitochondrial complex proteins (Fig 2B).

We next focused on the exosomal fraction isolated via sucrose gradient fractionation from the EV preparation, as described [10]. Compared with parental NSCs, the overall protein composition of exosomes by TMT-based proteomic analysis was similar to EVs (Fig 3A, S1 Data). However, we also found that several proteins were selectively depleted in exosomes by the additional purification step versus EVs (Fig 3B). Exosomes were also significantly enriched in proteins with GOCC annotations indicating localisation to the mitochondrial outer membrane, inner membrane, and matrix compared to parental NSCs (Fig 3C). Immunoblot analysis confirmed that fractions 6 to 9 corresponding to the expected exosomal density (i.e., between 1.13 and 1.20 g/ml) [10] were all enriched in mitochondrial complex proteins (Fig 3D). When looking at the DNA content of single fractions, we identified the mitochondrial gene *mt-ND1*—but not the *Sdhd* gene (Fig 3E), which unambiguously confirms that NSC exosomes, as well as EVs, harbour mitochondrial proteins and mtDNA. Altogether, these data suggest that mitochondrial proteins are found in EVs, irrespective of the protocols used for vesicle isolation from tissue culture media, which indicates the presence of either mitochondrial fragments or intact mitochondria shed by NSCs.

## Structurally and functionally intact mitochondria are found in EV preparations

We then further characterised NSC EVs using complementary biophysical and morphological approaches. Using tunable resistive pulse sensing (TRPS) analysis and nanoparticle tracking analysis (NTA), we found that EVs had a mode diameter ranging between 80 nm and 150.8 nm, respectively, as previously described [10] (S2 Fig). Size distribution was further investigated with a morphological analysis based on transmission electron microscopy (TEM). Among a heterogeneous population of EVs with a mean diameter of 350.13 nm (± 25.70 nm), we found that 27.81% (± 3.54) of particles corresponded to mitochondria-like structures with a mean diameter of 695.4 nm (± 68.47 nm) (Fig 4A). This finding was compatible with the presence of intact mitochondria in the EV preparation [24].

We next compared via nano flow cytometry (NanoFCM) analysis [26,27] our EV preparation [10] versus EVs$^{Mito\_depl.}$ [25]. We found that among the total EVs sized between 0.04 and 1 μm, 33.52% (Q2+Q3) were positive for the mitochondrial dye MitoTracker red (Fig 4B). Among these, 77.6% (Q2/[Q2+Q3]) also expressed the canonical EV marker CD63 (Fig 4B). On the contrary, MitoTracker red$^+$ mitochondria were almost completely lacking from EVs$^{Mito\_depl.}$, which most likely included only mitochondrial proteins and/or fragments.

We next used cryo-TEM [28] combined with immuno-gold labelling using antibody conjugated to gold nanoparticles (NP) to identify the presence of mitochondria within the EV

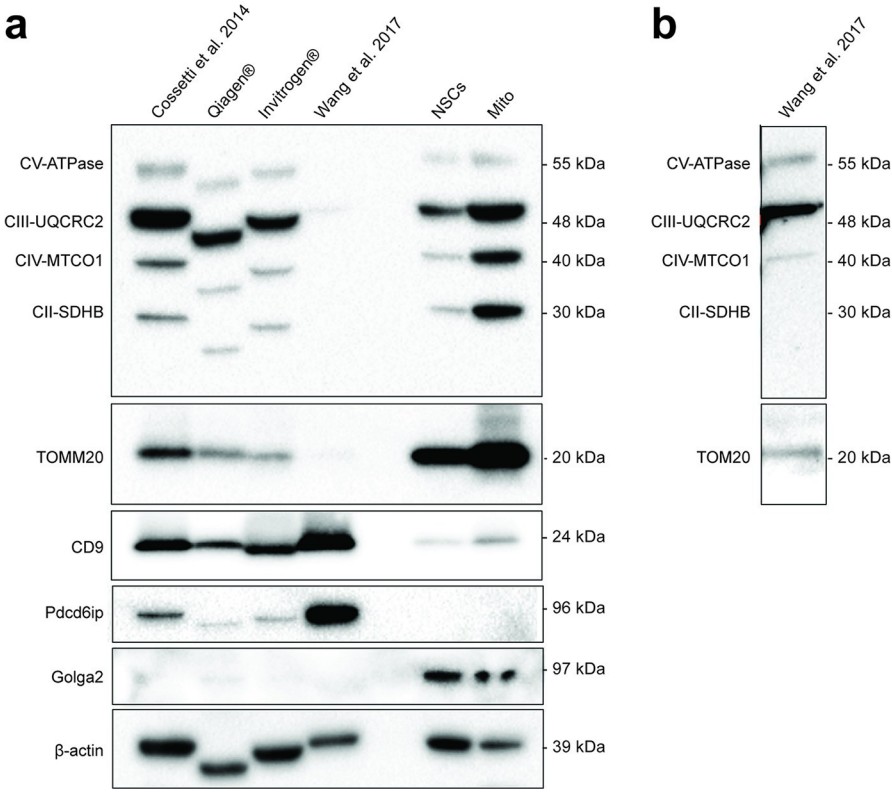

**Fig 2. Quality controls of EVs. (a)** Protein expression by WB analysis of EVs isolated using in house protocol, with commercial kits (Qiagen cat. No 76743 and Invitrogen cat. No 4478359), and an alternative protocol with an additional 0.22 μm ultrafiltration step. NSCs and isolated mitochondria (Mito) are used as comparative controls. Mitochondrial complexes proteins (CV-ATPase, CII-SDHA, CII-SDHB, CIV-MTCO1, and CIII-UQCRC2), mitochondrial outer membrane translocase (TOMM20), and exosomal positive (Pdcd6ip and CD9) and negative (Golga2) markers are shown, as well as β-actin. **(b)** Longer exposure of the lane containing the EVs isolated with the alternative protocol with an additional ultrafiltration step showing mitochondrial proteins. EV, extracellular vesicle; NSC, neural stem cell; WB, western blot.

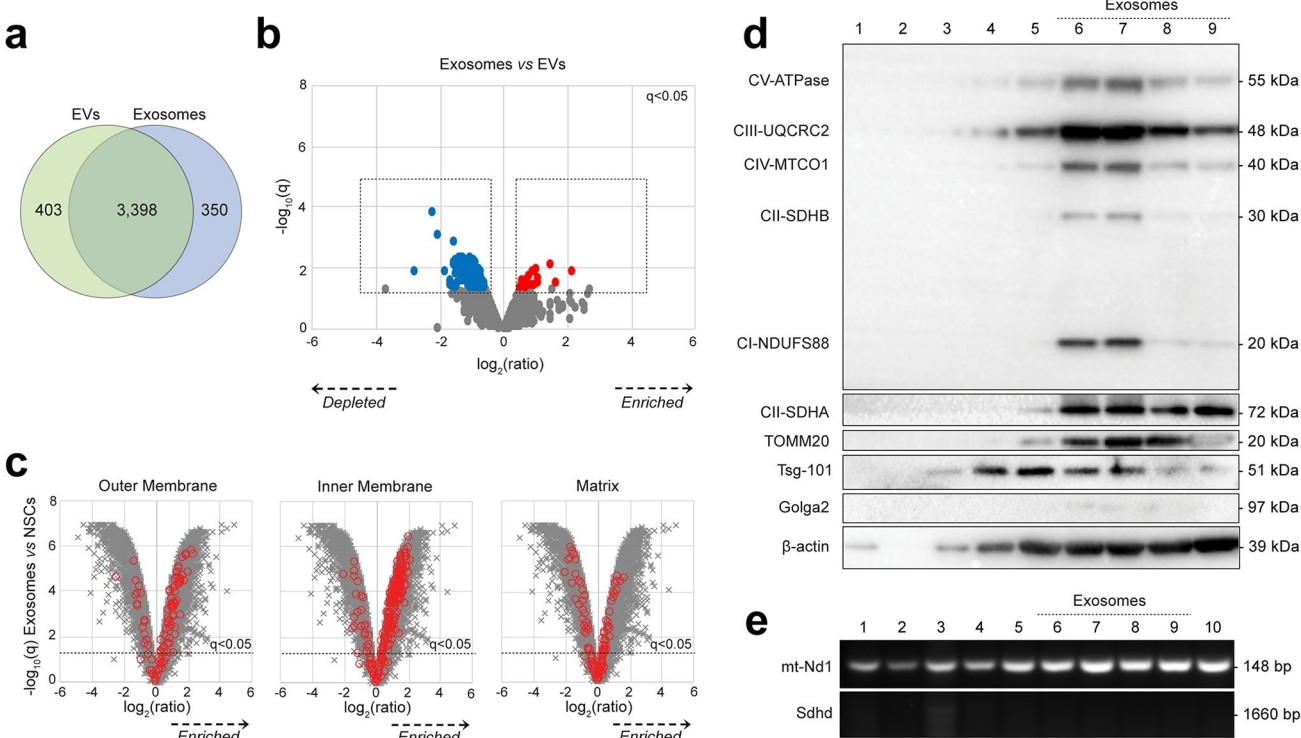

**Fig 3. Proteomic analysis of EVs and exosomes.** (a) Overview of proteins enriched in EVs and/or exosomes. Venn diagram shows overlap of 4,151/ 9,971 cellular proteins significantly enriched (q < 0.05) in either EVs (green) or exosomes (blue) or both, compared with NSCs. (Data available on ProteomeXchange, identifier PXD024368, and in S3 Data). (b) Relative abundance of proteins in exosomes (fractions 6–9) compared with EVs. Volcano plot shows statistical significance (*y* axis) vs. fold change (*x* axis) for 9,951/9,971 cellular proteins quantitated across all *N* = 3 biological replicates (no missing values) in the multiplex TMT-based functional proteomic experiment illustrated in Fig 1A. A total of 187 proteins were found to be significantly depleted in exosomes vs. EVs (blue), while 25 proteins were significantly enriched (red); q< 0.05 (S1 Data). (Data available on ProteomeXchange, identifier PXD024368, and in S3 Data). (c) Relative abundance of proteins from different mitochondrial compartments (outer membrane, matrix, and inner membrane) in exosomes (fractions 6–9) compared with NSCs. Volcano plots show statistical significance (*y* axis) vs. fold change (*x* axis) for 9,951/ 9,971 cellular proteins quantitated across all 3 biological replicates (no missing values) in the multiplex TMT-based functional proteomic experiment illustrated in Fig 1A. Proteins annotated with the following GOCC subcellular localisations are highlighted in red: mitochondrial outer membrane (GO:0005741, enriched in EVs); mitochondrial matrix (GO:0005759, enriched in EVs); and mitochondrial inner membrane (GO: 0005743, enriched in EVs). An FDR threshold of 5% is indicated (proteins with Benjamini–Hochberg FDR-adjusted *p*-values (q values) <0.05). (d) Representative protein expression analysis by WB of EV fractions (2–10) obtained via continuous sucrose gradient. Fractions 6–9 (corresponding to the expected exosomal density between 1.13 and 1.21 g/ml) were specifically enriched for mitochondrial complex proteins (CV-ATPase, CII-SDHA, CI-NDUFS88, CII-SDHB, CIV-MTCO1, and CIII-UQCRC2), for the mitochondrial outer membrane translocase TOMM20, and the exosomal marker Tsg-101 (while they were negative for the Golgi marker Golga2). β-actin is also shown. (e) Representative PCR amplification of DNA extracted from EV fractions (2–10) obtained via continuous sucrose gradient. The mitochondrial encoded gene *mt-ND1* was found to be present in most of the EV fractions, while the nuclear encoded mitochondrial gene *Sdhd* was used as negative control. EV, extracellular vesicle; FDR, false discovery rate; *mt-ND1*, mitochondrial gene NADH dehydrogenase subunit 1; PCR, polymerase chain reaction; *Sdhd*, succinate dehydrogenase complex subunit D; TMT, Tandem Mass Tag; WB, western blot.

preparations. TOMM20+ free mitochondria were found in all our crude EV preparations (Fig 4C) and, when saponin was used as a mild detergent to increase antibody permeability, we also identified TOMM20 and CD63 double positive particles with 3 distinct membranes (Fig 4D). Altogether, these approaches confirm that NSCs spontaneously release EVs in the submicron range in vitro, which include both free and encapsulated mitochondria.

We finally tested the functionality of mitochondria in the crude EV preparations by analysing the activity of the electron transport chain (ETC) in maintaining a mitochondrial transmembrane potential and respiration using a JC1 assay and high-resolution respirometry (HRR) [29], respectively. Contrary to EVs^Mito_depl., crude EV preparations showed a conserved mitochondrial membrane potential, which was responsive to the mitochondrial uncoupler

carbonyl cyanide *m*-chlorophenyl hydrazone (CCCP) (Fig 4E). In addition, EVs exhibited oxygen consumption when the substrates for the mitochondrial complexes were added to the EV preparation (Fig 4F and 4G). When we tested the activity of the mitochondrial respiratory chain complexes using blue native polyacrylamide gel electrophoresis (BN-PAGE) [30], intact protein complexes were isolated from EVs in native conditions (Fig 4H). This finding was coupled with a conserved catalytic activity of CI-CII-CIV, which suggests the presence of functionally intact mitochondrial complexes (Fig 4I) [31].

Altogether, these findings demonstrate that NSC EVs harbour a functional mitochondrial ETC and the potential for oxidative phosphorylation (OXPHOS).

## EV-associated mitochondria revert the auxotrophy of mtDNA-deficient cells

To investigate whether EVs have any effect on target cells, we first generated NSCs that constitutively express the mitochondrial MitoDsRed fluorescent reporter (MitoDsRed+ NSCs) that stably labels intact mitochondria in EVs [32,33]. In fact, the mitochondrial target sequence of MitoDsRed guarantees that the dsRed protein is accumulated only in those mitochondria conserving an intact membrane, thus enhancing the specificity of this tag [34].

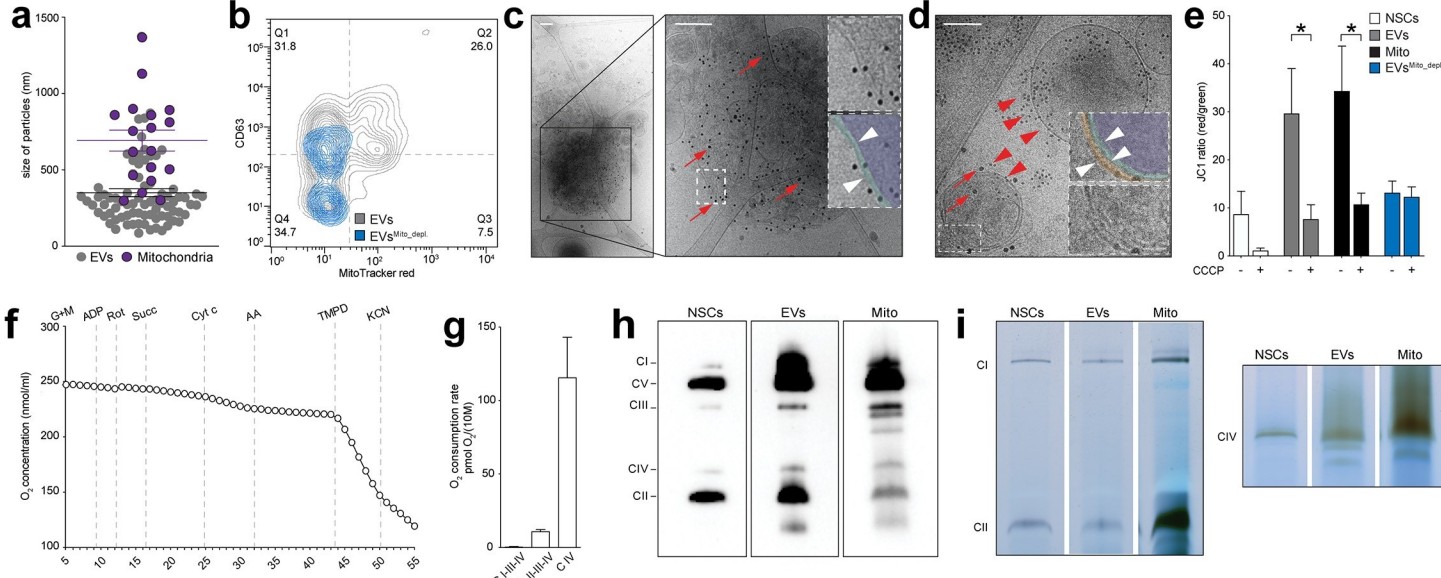

**Fig 4. NSC EVs include structurally and functionally intact mitochondria. (a)** TEM data showing size particle analysis and quantification of mitochondria found in the EV preparations (purple dots) compared to non-mitochondrial EVs (grey dots). Data are mean values (± SEM) from *N* = 2 biological replicates. (Data available in S3 Data). **(b)** Representative NanoFCM density plot of EVs and EVs^Mito_depl. labelled with the canonical EV marker CD63 and the mitochondrial dye MitoTracker red. (Data available in S3 Data). **(c)** Representative cryo-TEM image of a free mitochondria labelled with anti-TOMM20 (red arrows) antibody conjugated to 10-nm gold NP. Inset: magnified ROI pseudocolored (or not) to highlight the 2 mitochondrial membranes (white arrowheads). Scale bars: 200 nm. **(d)** Representative cryo-TEM image of NSC EVs treated with saponin and labelled with anti-CD63 (red arrowheads) and anti-TOMM20 (red arrows) antibodies conjugated to 10 nm and 20 nm gold NP, respectively. Inset: magnified ROI pseudocolored (or not) to highlight the 3 membranes (white arrowheads). Scale bar: 200 nm. **(e)** Mitochondrial membrane potential of NSCs, EVs, and Mito preparations treated (or not) with the mitochondrial uncoupler CCCP. *$p \leq 0.05$. Data are mean values (± SEM) from $N \geq 2$ independent experiments. (Data available in S3 Data). **(f)** Representative mitochondrial respiration of permealised EVs detected by HRR. Representative plot showing O₂ concentration changes over time upon serial additions of selected mitochondrial complexes substrates, inhibitors, and uncouplers, including CI substrates (G+M: ADP), CI inhibitor (rotenone: Rot), CII substrate (succinate: Succ), cytochrome *c* (Cyt c) to compensate for a possible loss due to outer membrane disruption, CIII inhibitor (antimycin A: AA), CIV electron donor (*N*,*N*,*N*′,*N*′-tetramethyl-*p*-phenylenediamine: TMPD) and CIV inhibitor (potassium cyanide: KCN). (Data available in S3 Data). **(g)** Complex respiratory rate in EVs. Data are mean values (± SEM) from *N* = 3 independent experiments. (Data available in S3 Data). **(h)** Representative image of mitochondria respiratory chain native complexes separated by BN-PAGE showing the presence of structurally intact respiratory complexes (CI–V) in NSCs, released EVs, and isolated Mito preparations. **(i)** Representative image of in situ gel activity of CI-II-IV in NSCs, EVs, and Mito obtained from BN-PAGE gel incubation for 24 hours. ADP, adenosine diphosphate; BN-PAGE, blue native polyacrylamide gel electrophoresis; CCCP, carbonyl cyanide m-chlorophenyl hydrazone; cryo-TEM, cryo-transmission electron microscopy; EV, extracellular vesicle; G+M, glutamate and malate; HRR, high-resolution respirometry; NanoFCM, nano flow cytometry; NP, nanoparticles; NSC, neural stem cell; ROI, region of interest; TEM, transmission electron microscopy.

We then used MitoDsRed[+] NSCs to generate MitoDsRed[+]-EVs and treat cells that had been depleted of mtDNA using extended low-dose ethidium bromide treatment [35] (Fig 5A). These L929 Rho[0] cells have an auxotrophic growth and dependence on extracellular uridine, which satisfies their energy demands despite the inhibition of dihydroorotate dehydrogenase (DHODH) allowing for cell survival [36].

In conditions of uridine deprivation, we found that L929 Rho[0] cells efficiently incorporated MitoDsRed[+]-EVs within 24 hours from treatment compared to L929 Rho[0] cells treated with a preparation enriched with isolated mitochondria (MitoDsRed[+]-Mito) (Fig 5B). At 5 days posttreatment, while only a minority of untreated L929 Rho[0] cells survived uridine depletion, L929 Rho[0] cells treated with MitoDsRed[+]-EVs displayed a significantly higher survival (Fig 5C). Finally, at 16 days posttreatment, we were able to sequence the *mt-ND3* mitochondrial gene from L929 Rho[0] cells treated with MitoDsRed[+]-EVs (Fig 5D), which indicates the efficient integration of (exogenous) EV-derived mtDNA in target cells and a correction of their intrinsic mitochondrial DNA dysfunction.

These results show that EV-associated mitochondria are incorporated by and restore the mitochondrial function of persistently mtDNA-depleted target cells.

## EV-associated mitochondria integrate into the mitochondrial network of mononuclear phagocytes

Mitochondrial function and immune metabolism guide the activation of mononuclear phagocytes in response to inflammatory stimuli [9]. Our recent work suggests that exogenous NSC

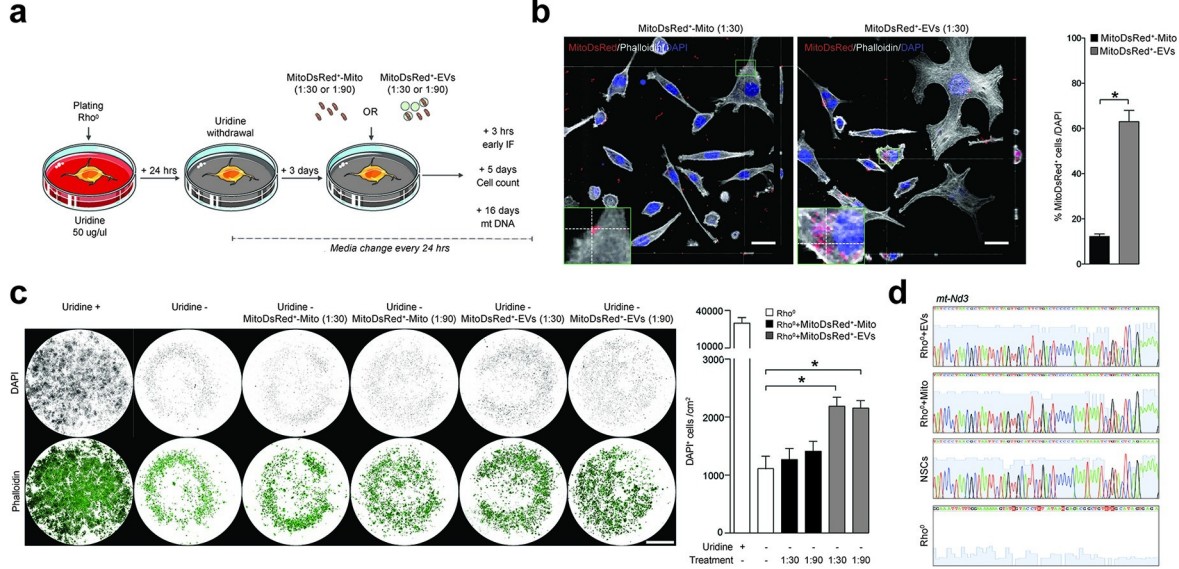

**Fig 5. EV-associated mitochondria revert the auxotrophy of mtDNA-depleted cells. (a)** Experimental setup for in vitro studies with L929 Rho[0] cells. L929 Rho[0] cells were deprived of uridine (Uridine[−]) and then treated with either MitoDsRed[+] EVs (ratio 1:30 or 1:90) or MitoDsRed[+] Mito (ratio 1:30 or 1:90) after 3 days. **(b)** Representative confocal images and quantification of Uridine[−] L929 Rho[0] cells showing incorporation of MitoDsRed[+] mitochondria at 24 hours from treatment with either EVs or Mito (ratio 1:30). Orthogonal section (XY) of Z-stacks is shown. Data are mean values (± SEM) from Data are from $N = 4$ biological replicates per condition (Mann–Whitney). *$p < 0.05$. Scale bars: 25 μm. (Data available in S3 Data). **(c)** Representative images and quantification of L929 Rho[0] cells surviving 5 days after treatment with either EVs or Mito. Data are mean values (± SEM) from $N = 4$ biological replicates per condition. *$p \leq 0.05$ (One-Way ANOVA, followed by Mann–Whitney). Scale bars: 3.25 mm. (Data available in S3 Data). **(d)** Sanger sequencing chromatograms showing the mitochondrial encoded gene *mt-ND3* in L929 Rho[0] cells at 16 days after treatment with either EVs or Mito. NSCs and L929 Rho[0] cells are used as positive and negative controls, respectively. EV, extracellular vesicle; mt-ND1, mitochondrial gene NADH dehydrogenase subunit 1.

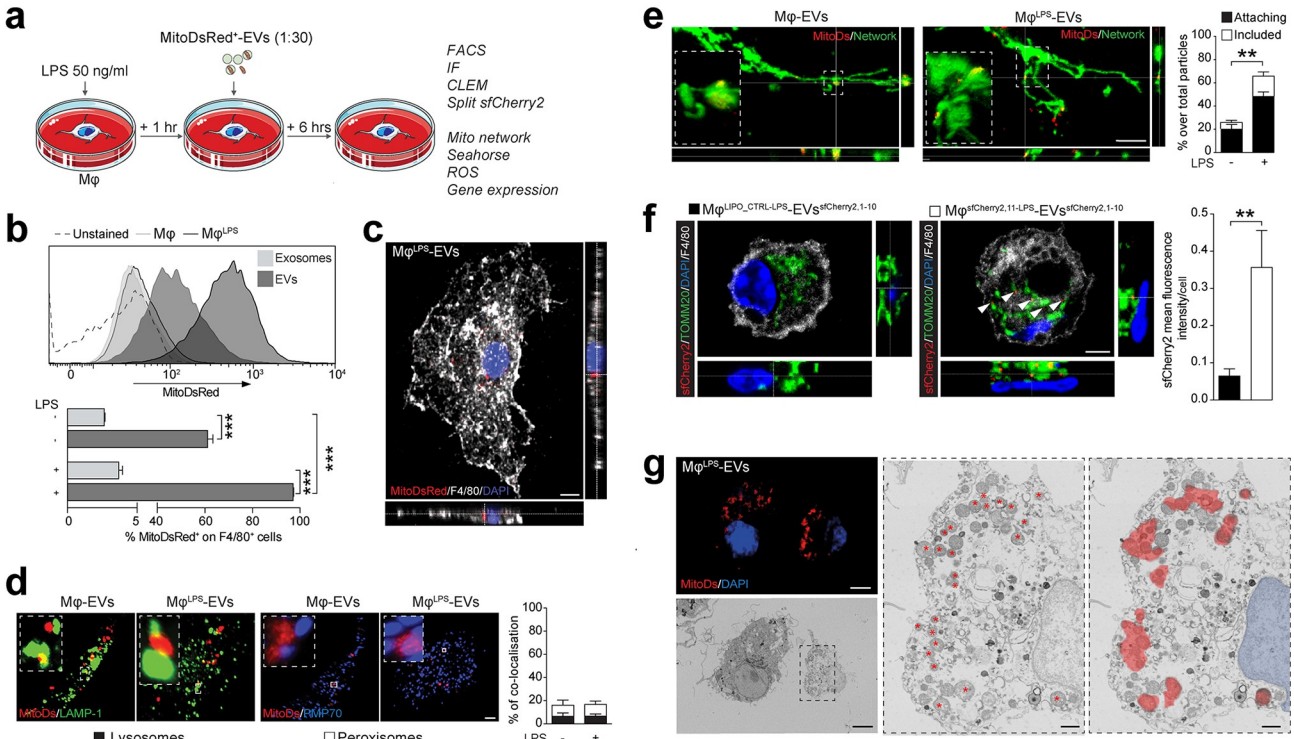

**Fig 6. EV-associated mitochondria integrate in the host mitochondrial network. (a)** Experimental setup for the functional in vitro studies of EVs on Mφ. Mφ$^{LPS}$ were treated with EVs spontaneously released from MitoDsRed$^+$ NSCs (ratio 1:30). Uptake of MitoDSred$^+$ particles and functional analyses of Mφ were assessed at 6 hours posttreatment. **(b)** Flow cytometry–based representative density plots of Mφ and Mφ$^{LPS}$ at 6 hours after treatment with MitoDsRed$^+$ EVs or exosomes. Data are mean % (± SEM). $^{***}p \leq 0.001$. $N = 3$ independent biological replicates. (Data available in S3 Data). **(c)** A representative confocal image (orthogonal section (XY) of Z-stacks is shown) of Mφ$^{LPS}$ treated with EVs at 6 hours, showing uptake of MitoDSred$^+$ mitochondria. Nucleus is stained with DAPI (blue). Scale bar: 3 μm. **(d)** Representative spinning disk micrographs (maximum intensity projection of Z-stacks) and quantification of MitoDsRed$^+$ EVs (red) co-localising with the lysosomal marker LAMP1 (green) or the peroxisomal marker PMP70 (blue) in Mφ. Data are mean values (± SEM). $^*p < 0.05$. $N \geq 10$ cells per condition (2 independent experiments). Scale bars: 2 μm. (Data available in S3 Data). **(e)** Representative spinning disk images (orthogonal section (XY) of Z-stacks is shown) and relative quantification showing MitoDSred$^+$ EVs (red) attached or included in the mitochondrial network of Mφ (previously stained with MitoTracker Green FM). Inset: magnified 3D surface reconstruction of included mitochondria (Imaris Software). Data are percentage of either attaching or including particles over total MitoDSred$^+$ particles in Mφ (± SEM). $^{**}p < 0.01$. $N \geq 5$ cells per condition from $N = 3$ independent experiments. Scale bars: 5 μm. (Data available in S3 Data). **(f)** Representative images (orthogonal section (XY) of Z-stacks) of a split FPs showing EVs$^{sfCherry2,1–10}$ fusing with Mφ$^{sfCherry2,11}$ (in red) juxtaposed to the host TOMM20$^+$ mitochondrial network (in green) at 6 hours from EV treatment. Nuclei are stained with DAPI (blue). Data are mean values (± SEM). $^{**}p < 0.01$. $N = 10$ cells per condition. Scale bars: 5 μm. (Data available in S3 Data). **(g)** Representative CLEM image of Mφ$^{LPS}$ treated with MitoDsRed$^+$-EVs for 6 hours. Top left panel, confocal image (orthogonal section (XY) of 1 Z-stack) showing MitoDsRed$^+$ EVs (red) and Mφ$^{LPS}$ nuclei (blue); bottom left panel, scanning EM image of the Mφ$^{LPS}$ depicted in the confocal image. Scale bars: 5 μm. Middle panel, magnified ROI of the Mφ$^{LPS}$ mitochondrial network ultrastructure; right panel, superposition of confocal and scanning EM images showing co-localisation of the MitoDsRed$^+$ mitochondria (red) with the host Mφ$^{LPS}$ mitochondrial network. Nucleus is pseudocolored in blue. Asterisks show MitoDsRed$^+$ mitochondria. Scale bars: 1 μm. CLEM, correlative-light electron microscopy; DAPI, 4′,6-diamidino-2-phenylindole; EV, extracellular vesicle; LPS, lipopolysaccharide; ROI, region of interest.

transplants have immunomodulatory functions and inhibit the activation of pro-inflammatory mononuclear phagocytes in response to endogenous metabolic signals in vivo [37]. As such, to gain further insights into the role of EVs in the immunomodulatory effects of NSCs, we next investigated whether these are trafficked to mononuclear phagocytes and, in so doing, affect their function.

Bone marrow–derived Mφ were challenged with lipopolysaccharide (LPS) to generate reactive, pro-inflammatory macrophages (Mφ$^{LPS}$) and then treated with MitoDsRed$^+$-EVs or exosomes (Fig 6A). Mφ$^{LPS}$ showed the highest intracellular MitoDsRed$^+$ positivity via fluorescent-activated cell sorting (FACS) analysis [97.17% (± 0.20)] at 6 hours after treatment, compared

to either resting Mφ [61.23% (± 2.15)] or Mφ$^{LPS}$ treated with exosomes [3.7% (± 0.23)] (Fig 6B). This finding suggests the incorporation of mitochondria, which is dependent on Mφ activation state and is predominantly mediated by crude EVs rather than the exosomal fraction (Fig 6C).

We next proceeded to investigate the intracellular fate of exogenous mitochondria in Mφ treated with EVs via co-localisation analysis of confocal high-resolution spinning disk images. We found that 6.58% (± 2.00) and 10.14% (± 2.85) of MitoDsRed$^+$ mitochondria co-localised with either the lysosomal (LAMP1) or the peroxisomal (PMP70) markers (Fig 6D), respectively. These findings suggest limited trafficking of MitoDsRed$^+$ mitochondria to these cellular compartments. Instead, 48.24% (± 4.43) and 17.75% (± 3.56) of MitoDsRed$^+$ mitochondria were found either attached or included within the host Mφ$^{LPS}$ mitochondrial network (Fig 6E, S1 Video).

To further address the possibility of fusion and incorporation of exogenous mitochondria into the endogenous mitochondrial network of Mφ, we first used split self-associating fluorescent proteins (FPs) to detect EVs to Mφ mitochondria contacts [38]. Mφ transiently expressing the sfCherry2$_{11}$ protein fused with the mitochondrial protein TOMM20 (i.e., Mφ$^{sfCherry2,11}$) were stimulated with LPS and treated with EVs derived from NSCs transiently expressing the sfCherry2$_{1-10}$ protein (i.e., EVs$^{sfCherry2,1-10}$) [38]. Fluorescent signal of the sfCherry2 FP was found in juxtaposition to the Mφ$^{LPS}$ endogenous mitochondrial network, suggesting fusion of EVs with the host mitochondrial network (Fig 6F). Then, we performed a correlative-light electron microscopy (CLEM) experiment on Mφ$^{LPS}$ treated with MitoDsRed$^+$-EVs to study particle incorporation by combining immunofluorescence labelling with high-resolution contextual ultrastructure. As final direct confirmation of the above, MitoDsRed$^+$ signal fully colocalised with ultrastructurally defined mitochondria integrated in the host Mφ$^{LPS}$ mitochondrial network (Fig 6G).

Thus, we show that the majority of EV-associated mitochondria preferentially escape the lysosomal and peroxisomal pathways in pro-inflammatory Mφ and instead co-localise and fuse with the endogenous mitochondrial network.

## Pro-inflammatory mononuclear phagocytes uptake EV-associated mitochondria via endocytosis

While we have previously described evidence of EV incorporation in target cells via fusion with the plasma membrane [10], the mechanisms driving EV uptake by activated mononuclear phagocytes are not yet fully understood. As such, to investigate the incorporation of EVs into Mφ$^{LPS}$, we pretreated Mφ with either the (actin-mediated) phagocytosis and endocytosis inhibitor cytochalasin D (Cyto) [39,40] or the (dynamin and clathrin-mediated) endocytosis only inhibitors Dynasore and Pitstop 2 (D/P) [41,42] (Fig 7A).

Following a 30-minute pretreatment with the selected inhibitors, Mφ were treated with MitoDsRed$^+$-EVs. Mφ$^{LPS}$ showed a significantly enhanced incorporation of MitoDsRed$^+$-EVs as early as 15 minutes after exposure, which was completely blocked by pretreatment of Mφ$^{LPS}$ with the phagocytosis inhibitor Cyto or the endocytic pathways inhibitors D/P (Fig 7B and 7C).

Altogether, these findings suggest that NSC EVs are predominantly incorporated via endocytosis in activated Mφ and trafficked into the host mitochondrial network.

## EV-associated mitochondria inhibit the metabolic switch of pro-inflammatory mononuclear phagocytes

During inflammation, Mφ undergo major changes in their function and metabolism, which are associated with modifications of their mitochondrial network dynamics [43,44]. As such,

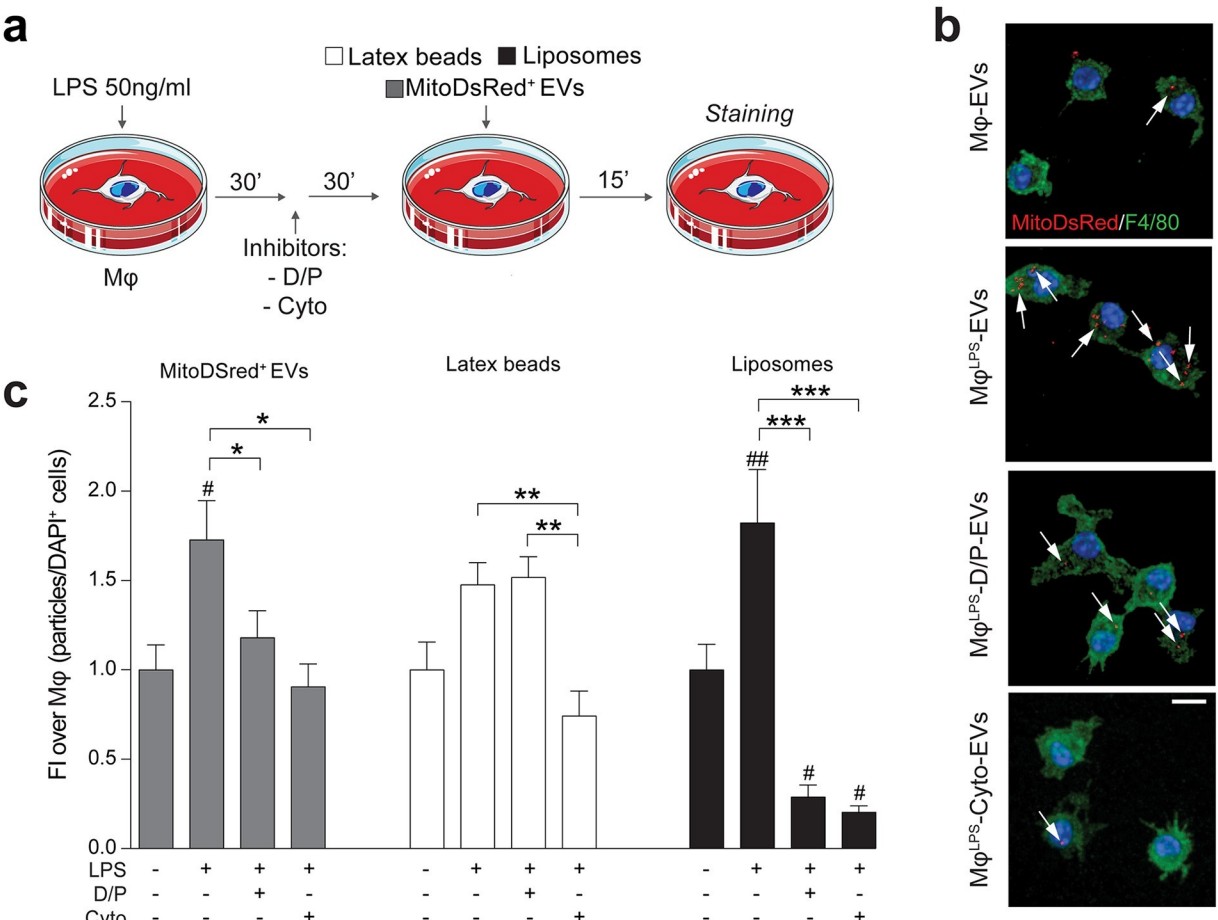

**Fig 7. Pro-inflammatory mononuclear phagocytes uptake EV-associated mitochondria via endocytosis. (a)** In vitro experimental setup of EV uptake studies in Mφ$^{LPS}$. Mφ$^{LPS}$ were treated with either Cyto or D/P and then exposed to MitoDsRed$^+$ EVs (1:30). Latex beads and liposomes were used as positive controls of phagocytosis and endocytosis, respectively. **(b, c)** Representative confocal microscopy images (maximum intensity projection of Z-stacks) and quantification of MitoDsRed$^+$ EV (red) uptake in Mφ$^{LPS}$ (stained for F4/80, green) in the presence or absence of endocytosis (D/P) and actin mediated phagocytosis/endocytosis (Cyto) inhibitors. Nuclei are stained with DAPI (blue). Data are mean FI over unstimulated Mφ (± SEM) from $N \geq 8$ ROIs per condition. $^{\#}p < 0.05$, $^{\#\#}p < 0.01$ vs. unstimulated Mφ. $^{*}p < 0.05$, $^{**}p < 0.01$, $^{***}p < 0.001$. Scale bars: 10 μm. (Data available in S3 Data). Cyto, Cytochalsin; D/P, Dynasore and Pitstop 2; DAPI, 4′,6-diamidino-2-phenylindole; EV, extracellular vesicle; FI, fold induction; LPS, lipopolysaccharide; ROI, region of interest.

we next investigated the structure of the mitochondrial network of pro-inflammatory Mφ after EV treatment. We found that, while the stimulation with LPS promoted mitochondrial fission, the uptake of EVs and the integration of exogenous mitochondria into the host mitochondrial network led to a significant increase in fused mitochondria as early as 6 hours after treatment (Fig 8A).

To understand the relevance of these structural changes and their functional consequences in a broader context, we analysed the gene expression profiles of Mφ$^{LPS}$ treated with EVs using RNA expression microarrays (S2 Data). Generally Applicable Gene-set Enrichment (GAGE) analysis [45] allowed us to identify specific Kyoto Encyclopedia of Genes and Genomes (KEGG) pathways up-regulated in Mφ$^{LPS}$ treated with EVs. Pathways related to ribosomes (mmu03010, q-value <0.01), carbon metabolism (mmu01200, q-value = 0.04), OXPHOS (mmu00190, q-value = 0.04), antigen processing and presentation (mmu04612, q-value = 0.04), and phagosomes (mmu04145, q-value = 0.04) were all up-regulated in Mφ$^{LPS}$ treated with EVs (Fig 8B).

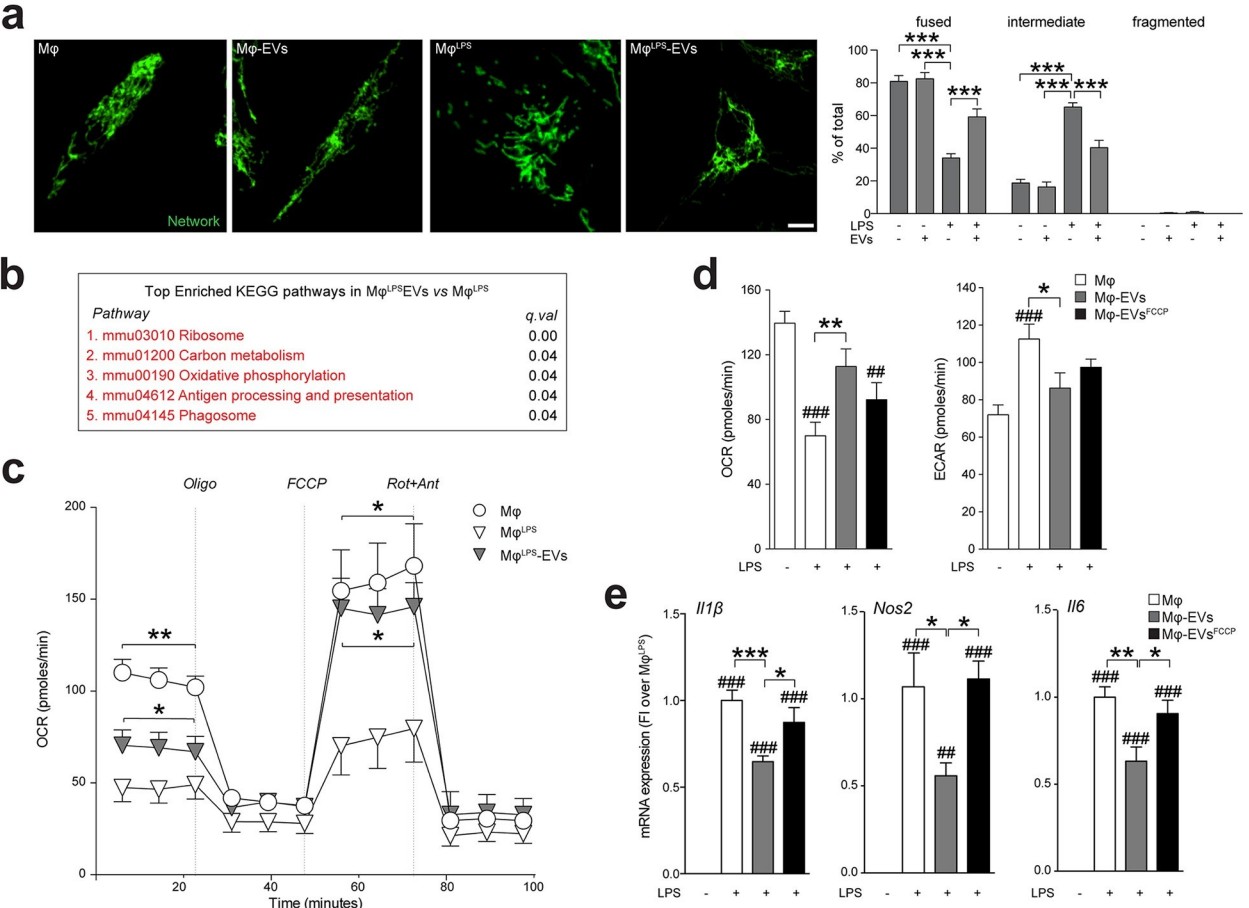

**Fig 8. The transfer of EV-associated mitochondria inhibits the metabolic switch of pro-inflammatory mononuclear phagocytes. (a)** Representative spinning disk images and quantification showing Mφ mitochondrial network labelled with TOMM20 (green) polymorphic dynamics after LPS stimulation and/or EV treatment (1:30). Data are expressed as mean % (± SEM). ***$p < 0.001$. $N = 6$ biological replicates. Scale bars: 4 μm. (Data available in S3 Data). **(b)** Top enriched KEGG pathways in genes up-regulated in EV-treated vs. untreated Mφ$^{LPS}$ at 6 hours. Expression data obtained by microarray analysis. (Data available on ArrayExpress, identifier E-MTAB-8250). **(c)** XF assay of the OCR during a mitochondrial stress protocol of Mφ$^{LPS}$ at 6 hours from EV treatment (1:30). Unstimulated Mφ were used as controls. Data are mean values (± SEM). *$p < 0.05$, **$p < 0.01$ vs. Mφ$^{LPS}$. $N = 2$ independent experiments. (Data available in S3 Data). **(d)** XF assay of the basal OCR and ECAR of Mφ$^{LPS}$ at 6 hours from treatment with EVs or treatment with EVs preexposed to the uncoupling agent FCCP vs. Mφ$^{LPS}$. Unstimulated Mφ were used as controls. Data are mean values (± SEM). ##$p < 0.01$, ###$p < 0.001$ vs. unstimulated Mφ. *$p < 0.05$, **$p < 0.01$. $N \geq 4$ technical replicates from $N \geq 2$ independent experiments. (Data available in S3 Data). **(e)** Expression levels (qRT-PCR) of pro-inflammatory genes (*Il1β*, *Nos2*, and *Il6*) in Mφ$^{LPS}$ at 6 hours from treatment with EVs or treatment with EVs preexposed to the uncoupling agent FCCP. Data are mean FI over unstimulated Mφ (± SEM). ##$p < 0.01$, ###$p < 0.001$ vs. unstimulated Mφ. *$p < 0.05$, **$p < 0.01$, ***$p < 0.001$. $N \geq 3$ biological replicates from $N \geq 2$ independent experiments. (Data available in S3 Data). ECAR, extracellular acidification rate; EV, extracellular vesicle; FI, fold induction; KEGG, Kyoto Encyclopedia of Genes and Genomes; LPS, lipopolysaccharide; OCR, oxygen consumption rate; qRT-PCR, quantitative real-time polymerase chain reaction; XF, extracellular flux.

Interestingly, among the genes differentially expressed in the OXPHOS pathway, several genes encoding for the different subunits of the ETC were up-regulated in Mφ$^{LPS}$ after EV treatment, as shown by the Pathview diagram [46] (S3 Fig), which suggests a putative increase of cellular respiration.

As such, we next measured the oxygen consumption rate (OCR) of Mφ$^{LPS}$ and found that the basal OCR, as well as the maximum respiratory capacity, was significantly increased in Mφ$^{LPS}$ after EV treatment (Fig 8C), but not after treatment with either exosomes or EVs$^{Mito-depl.}$ (S4 Fig). These findings are in line with an increase in maximal respiration rate as an

index of metabolic activity associated with fused mitochondrial networks [47], and they show that EVs can revert the transient mitochondrial dysfunction associated with the pro-inflammatory state of Mφ.

To further prove that functional mitochondria trafficked within EVs were indeed responsible for the abovementioned changes, we treated Mφ$^{LPS}$ with EVs that had been preexposed to the mitochondrial un-coupler carbonyl cyanide-4-(trifluoromethoxy)phenylhydrazone (FCCP) (defined as EV$^{FCCP}$). While treatment with control EVs rescued the changes in the OCR and extracellular acidification rate (ECAR) induced by LPS in Mφ, treatment with EV$^{FCCP}$ failed to do so (Fig 8D). Moreover, while control EVs succeeded in down-regulating the expression of the LPS-induced pro-inflammatory cytokine genes *Il1b*, *Il6*, *and Nos2* in Mφ$^{LPS}$, this was not the case for Mφ$^{LPS}$ treated with EV$^{FCCP}$ (Fig 8E), exosomes, or EVs$^{Mito-depl.}$ (S4 Fig).

Overall, these data show that EV-associated mitochondria integrate into the transiently dysfunctional host mitochondrial network of pro-inflammatory Mφ, where they reestablish physiological mitochondrial dynamics, cellular metabolism, and reduce inflammatory gene profiles.

## Transplanted NSCs transfer mitochondria to host cells during EAE in vivo

Previous evidence has suggested that mitochondrial transfer occurs in vivo and may be involved in diverse pathophysiological situations, including CNS injury and cancer progression [12]. Therefore, in order to determine if our in vitro findings had any functional relevance in vivo, NSCs or EVs were injected ICV at the peak of disease (PD) into mice with MOG$_{35-55}$-induced chronic EAE, an animal model of multiple sclerosis.

To reliably detect NSCs and EVs in vivo, NSCs were previously transduced in vitro with both the MitoDsRed fluorescent reporter and farnesylated green fluorescent protein (fGFP) to generate fGFP$^{+}$/MitoDsRed$^{+}$ NSCs. EVs spontaneously released by fGFP$^{+}$/MitoDsRed$^{+}$ NSCs in vitro were then collected and used for in vivo studies (Fig 9A).

In line with published results, a single ICV injection of NSCs resulted in a significant amelioration of EAE disease severity when compared to PBS-injected EAE mice [48] (Fig 9B). Likewise, we found that a single ICV injection of EVs was able to significantly ameliorate EAE disability in mice compared to PBS-injected EAE mice. On the contrary, the ICV injection of either EVs$^{Mito\_depl.}$ or EVs$^{CD63\_depl.}$ failed to ameliorate the clinical deficits of EAE mice, compared to PBS-injected EAE mice.

Following the end of the clinical observation period (55 days post immunisation, dpi), brains were analysed to locate exogenous MitoDsRed$^{+}$ immunoreactivity within the host CNS tissue. In line with the expected turnover of the MitoDsRed protein [49], we could not identify any MitoDsRed$^{+}$ immunoreactivity in EAE mice injected with EVs. Rather, fGFP$^{-}$/MitoDsRed$^{+}$ mitochondria were found in both EAE mice and control nonimmunised mice close to cellular grafts, suggesting the local release and transfer of mitochondria from NSCs to host CNS cells.

To determine the target cell(s) of these mitochondrial transfer events, we next quantified the number of fGFP$^{-}$/MitoDsRed$^{+}$ mitochondria within the 3 major CNS cell types (astrocytes, neurons, and oligodendrocytes) and the immune cells that comprise most of the EAE inflammatory lesions (T cells and mononuclear phagocytes) (Fig 9C). Our analysis revealed that the majority of fGFP$^{-}$/MitoDsRed$^{+}$ mitochondria were predominantly transferred to F4/80$^{+}$ mononuclear phagocytes [52.5% (± 1.85)] and, to a lower extent, GFAP$^{+}$ astrocytes [38.25% (± 5.44)] during EAE (Fig 9C and 9D), compared to control, nonimmunised mice [31.0% (± 2.71) and 20.25% (± 7.97)] (Fig 9C). A minor fraction of fGFP$^{-}$/MitoDsRed$^{+}$ mitochondria

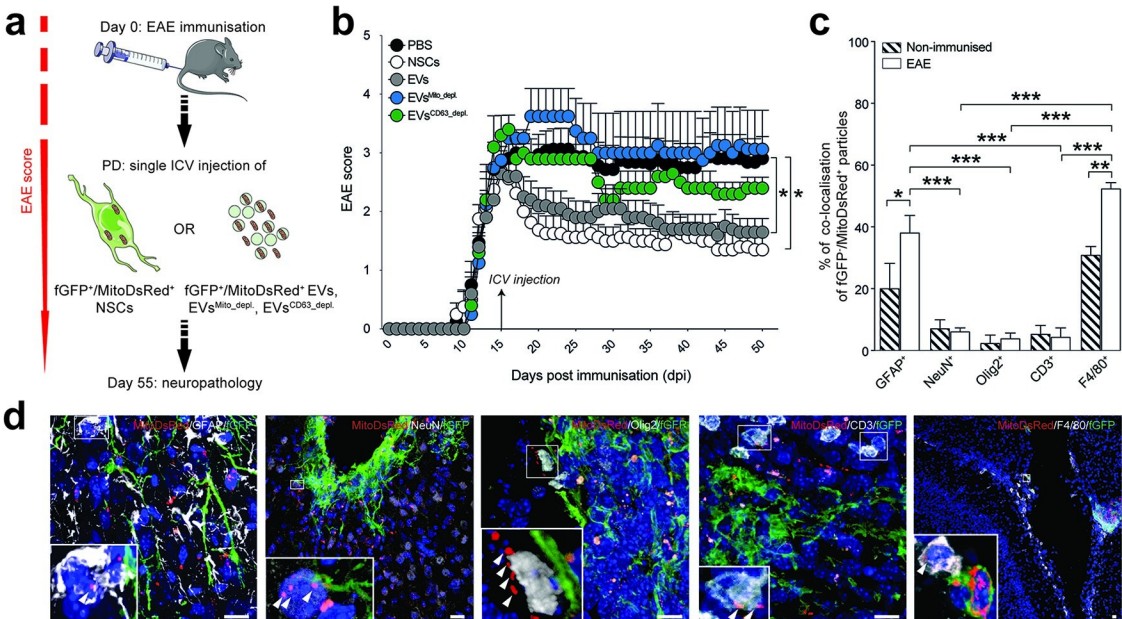

**Fig 9. Transplanted NSCs transfer mitochondria to mononuclear phagocytes and astrocytes during EAE in vivo. (a)** In vivo experimental setup of EV and NSC treatment in EAE mice. At PD, mice received a single ICV injection of either fGFP$^+$/MitoDsRed$^+$ NSCs, EVs derived from fGFP$^+$/MitoDsRed$^+$ NSCs (EVs), or fGFP$^+$/MitoDsRed$^+$ EVs depleted of mitochondria (EVs$^{Mito\_depl.}$), or fGFP$^+$/MitoDsRed$^+$ EVs depleted of CD63$^+$ EVs (EVs$^{CD63\_depl.}$). Behavioural analysis was carried out daily until the end of the experiment. Neuropathology was performed at 55 dpi. **(b)** Behavioural outcome showing significant amelioration of the EAE score in mice treated with EVs ($N = 5$) and NSCs ($N = 5$), but not in EAE mice treated with EVs$^{Mito\_depl.}$ ($N = 4$) or EVs$^{CD63\_depl.}$ ($N = 5$) vs. PBS ($N = 8$). Data are mean values ($\pm$ SEM). $^*p < 0.05$. (Data available in S3 Data). **(c)** Percentage of fGFP$^-$/MitoDsRed$^+$ particles co-localising with GFAP$^+$ astrocytes, NeuN$^+$ neurons, Olig2$^+$ oligodendrocytes, CD3$^+$ T cells, or F4/80$^+$ mononuclear phagocytes. fGFP$^+$/MitoDsRed$^+$ NSCs were injected ICV into EAE mice (white bars) and in nonimmunised control mice (hatched bars). Data are mean values ($\pm$ SEM) from $N = 4$ biological replicates. $^*p < 0.05$, $^{**}p < 0.01$, $^{***}p < 0.001$. (Data available in S3 Data). **(d)** Representative pictures of mitochondrial transfer events detected with confocal imaging (maximal projection). Transfer of MitoDsRed$^+$ particles (arrowheads) is shown between fGFP$^+$/MitoDsRed$^+$ NSCs and GFAP$^+$ astrocytes (cortex), NeuN$^+$ neurons (cortex), Olig2$^+$ oligodendrocytes (corpus callosum), CD3$^+$ T cells (meninges), or F4/80$^+$ mononuclear phagocytes (IV ventricle). Long processes of NSCs can be seen in green, while nuclei are stained with DAPI (blue). Scale bars: 20 μm. DAPI, 4′,6-diamidino-2-phenylindole; dpi, days post immunisation; EAE, experimental autoimmune encephalomyelitis; EV, extracellular vesicle; ICV, intracerebroventricular; NSC, neural stem cell; PD, peak of disease.

(approximately 15%) was distributed between NeuN$^+$ neurons, Olig2$^+$ oligodendrocytes, or CD3$^+$ T cells during EAE (Fig 9C and 9D).

These results show that mitochondrial transfer from NSCs happens in vivo and that it is modulated in conditions of neuroinflammation to be predominantly directed towards mononuclear phagocytes and host astrocytes.

## Discussion

Extracellular release of mitochondria and horizontal mitochondria transfer between cells are reported in several cells and organs, including the CNS [50,51]. However, the relevance and the biological function of these phenomena are still a matter of debate.

On the one hand, cells can release dysfunctional mitochondria for recycling and disposal [52,53]. Mitoptosis—the selective elimination of malfunctioning mitochondria—is described in cells under conditions of severe mitochondrial stress where the occlusion of mitochondrial clusters by a membrane ("mitoptotic body") allows its protrusion from the cell [54]. Similarly, transmitophagy—a process of transcellular degradation of damaged mitochondria through horizontal transfer—is observed between neurons and astrocytes [52] and between

mesenchymal stem cells (MSCs) and macrophages [53]. In addition, it has been shown that during oxidative stress, mitochondria produce their own mitochondria-derived vesicles (MDVs) that are trafficked intracellularly either to peroxisomes [55] or to multivesicular bodies and the exosomal pathway [56]. Overall, the effect of these mechanisms is to enhance the survival of the donor cell via disposal of dysfunctional mitochondria by routes that include unloading into neighbouring cells.

On the other hand, cells can also release intact mitochondria which retain functional properties [50,51,57–60]. The transfer of healthy mitochondria has been demonstrated in different tissues and organs where its main role is to maintain local homeostasis. In the CNS, astrocytes provide healthy mitochondria to damaged neurons to restore normal OXPHOS both in vitro and in vivo [50,51]. Similarly, endothelial progenitor cells support brain endothelial energetics and barrier integrity through extracellular mitochondrial transfer [57]. MSCs exchange mitochondria to foster cytoprotection in a variety of target cells (including cardiomyocytes, endothelial cells, and corneal epithelial cells) in vitro [58–60] and in vivo [58].

Herein, we first investigated the protein content of EVs that are spontaneously released by NSCs. We found that EVs were enriched in mitochondrial proteins of the outer membrane, matrix, inner membrane, and ETC. Subsequent biophysical (genomic polymerase chain reaction [PCR] and NanoFCM) and morphological (cryo-TEM) approaches confirmed the presence of intact mitochondria, either as free organelles or encapsulated in EVs, as previously shown in other cellular systems [18,19,61]. Most importantly, we demonstrate that mitochondria shed by NSCs via EVs have intact complexes, active complex activity, and conserved mitochondrial membrane potential and respiration. Altogether, these data show that NSCs release mitochondria into the extracellular space, wherein they still harbour functional properties that can be transferred to target cells.

Unravelling the physiological significance of mitochondrial transfer via EVs required the development of novel fluorescence and genetic mitochondrial tracking tools. To this aim, we generated NSC lines stably expressing the MitoDsRed protein, which allowed for the expression of this fluorescent reporter in NSC EVs. Thanks to this approach, we first provided evidence of functional mitochondria transfer from NSCs to L929 Rho$^0$ cells, where EVs succeeded in reverting their intrinsic mitochondrial dysfunction and auxotrophy.

Mitochondria transfer is emerging as a novel mechanism regulating the activity of the immune system. Besides the well-described release of mitochondria from immune cells such as monocytes [18,62], other immune regulatory cells modulate the activity of inflammatory cells via horizontal mitochondrial transfer [63]. As such, we next questioned whether EVs could also exert any regulatory functions on mononuclear phagocytes that display a transient dysfunction of mitochondria secondary to LPS stimulation.

Key to these investigations was to reveal how extracellular mitochondria enter into Mφ, which is an important step for the future development of treatment strategies designed to transfer healthy mitochondria from stem cells into immune cells.

Our experiments showed that Mφ$^{LPS}$ treated with EVs incorporated exogenous mitochondria preferentially via clathrin- or dynamin-mediated endocytosis. This is in line with previous reports showing that mitochondrial transfer events from MSCs to target cells are significantly reduced by endocytosis inhibition [60]. Most importantly, our data suggest that the uptake of EVs in target cells is not primarily mediated by phagocytosis but rather endocytic processes that may include micropinocytosis [64]. Indeed, we found that only a minority of the internalised EVs co-localised with either lysosomes or peroxisomes, while transferred mitochondria preferentially escaped the lysosomal/peroxisomal pathway. Instead, via complementary split FPs and CLEM experiments, we unambiguously show that EV-associated mitochondria fuse and integrate with the endogenous mitochondrial network of target Mφ$^{LPS}$ [63].

Integration of NSC EVs in target pro-inflammatory Mφ induced major changes in mitochondrial dynamics, gene expression profiles, and metabolism. EV treatment not only increased the number of fused mitochondria, but also induced an increase of genes related to OXPHOS, which was linked with an increase in both basal and maximal respiratory capacity of Mφ[LPS]. This is in line with previous evidence showing that inhibition of mitochondrial fission can reduce the glycolytic reprogramming of pro-inflammatory Mφ [65] and that OXPHOS-dependent ATP production can be restored by transfer of exogenous mitochondria [15,66,67].

It is interesting to note that the immunomodulatory effect of transferred mitochondria to immune cells reflects the activation status of their parental cells. While mitochondria derived from apoptotic cells are potent activators of innate immune responses, mitochondria derived from healthy cells are significantly less inflammatory [68]. Similarly, mitochondria from stressed monocytes, but not mitochondria from resting cells, induce type I interferon signalling in endothelial cells [62].

Here, we show that NSCs release both free and membrane encapsulated mitochondria with the capacity to restore OXPHOS and to reduce the pro-inflammatory gene profile of Mφ[LPS]. These effects are determined by the mitochondrial activity in parental cells rather than by the mere presence of mitochondrial content released, as shown by our experiments in which EVs treated with the mitochondrial uncoupler FCCP, or EVs depleted from mitochondria, failed to change the gene expression profile and metabolism of recipient Mφ[LPS]. Our findings are in line with data suggesting that disrupting electron transport (or ATP synthesis) in mitochondria significantly attenuates their protective transfer effect, implying that intact OXPHOS is indispensable for this function [69].

Since our observations were based solely on in vitro models, we decided to investigate the relevance of mitochondrial transfer from NSCs in vivo in a mouse model of neuroinflammation. Mice with EAE were treated with EVs or NSCs previously transduced with FPs to identify exogenous mitochondria and cellular grafts within the host CNS. While NSC EVs could not be reliably identified at the time point chosen for neuropathological analysis, EAE mice treated with NSCs showed exogenous mitochondria being exchanged between the graft and host cells, suggesting a continuous production and release of mitochondria by NSCs in vivo. Of note, the majority of the mitochondrial transfer events was observed between NSCs and mononuclear phagocytes, strengthening the relevance of our in vitro findings and suggesting that similar immunoregulatory effects may be relevant also in vivo.

Finally, we found that a single ICV injection of EVs—but not EVs depleted of mitochondria or the CD63-depleted EV fraction—induced a significant amelioration of clinical deficits in EAE mice to a level that is comparable to that of parental NSCs. While it is not possible to fully dissect the functional impact of free or EV-encapsulated mitochondria separately, our data strongly suggest that mitochondria in EV preparations, and the CD63[+] EV fraction, are indispensable for the in vivo therapeutic effect. In addition to classical cellular grafts [37], these new data further support the use of EV-based acellular therapies in regenerative neuroimmunology [70].

Our work provides direct proof of evidence that horizontal transfer of functional mitochondrial occurs for NSCs, paving the way for future investigations aiming at discovering the intracellular signals that link upstream mitochondrial release to downstream regulation of target cell phenotype and function [71]. Moreover, while previous data showed that direct exposure to purified mitochondria activates leukocytes [72], herein, our findings support the hypothesis that NSCs mitochondrial horizontal transfer is instead a mechanism of cell-to-cell signalling that reduces Mφ[LPS] activation. Future therapeutic implications of this research will need to consider strategies to pharmacologically enhance intercellular organelle transfer when

desirable, such as in neuroinflammation [9,73,74], or block its occurrence when it is deleterious [75].

Nonetheless, we are also fully aware of the main limitations of our work. First, technical issues in the isolation of EVs and different nomenclatures of extracellular particles have to be considered. In this sense, a recent study by Jeppesen and colleagues suggests that much of the protein components of classical exosomes are absent when EVs are isolated and characterised using high-resolution density gradient fractionation and direct immunoaffinity capture [76]. However, several of the protocols used in this paper involved prolonged incubation and resuspension of immunoaffinity beads in LDS buffer, possibly causing an artefact of innate protease digestion of the EV protein cargo (including proteins that are not embedded in membranes or protected by glycosylation). Furthermore, a limited number of cell types were examined in the Jeppesen study. As such, the need for a reassessment of exosome composition and a better framework for the distinction of EVs and non-vesicular fractions is still needed.

A second potential caveat of our study is the unknown nature of the NSC mitochondrial release mechanism, which is, however, out of the main scope of this work. In fact, indirect evidence suggests that the release of mitochondria is correlated with the metabolic state of the donor cell, as the same cell derived from different tissue sources has different mitochondrial donor properties, which are correlated with its respiratory state [77]. Cells with high mitochondrial respiration capacities are associated with lower mitochondrial transfer, which is in theory compatible with a model where donor cells with the least basal dependence on mitochondrial function may have increased mitochondrial transfer abilities. Interestingly, NSCs are highly glycolytic cells, which rely on glycolysis rather than OXPHOS for energy production [78]. As such, it is not unrealistic to speculate that NSCs might have increased mitochondrial donor properties compared to other cell types and that specific stimuli affecting their metabolism might further enhance their own mitochondrial transfer. Mechanistically, our data suggest that functionally intact mitochondria are released by NSCs via EVs/ectosomes rather than canonical exosomes. However, we cannot exclude that the prolonged isolation procedure used to isolate exosomes might have impacted on their functionality. Further experiments will be needed to understand the mechanisms regulating mitochondria release from NSCs and the intracellular pathways involved.

In conclusion, our work provides new insights to the contribution of mitochondria to the content and biological activity of EVs released by NSCs, suggesting that EV-mediated paracrine actions and mitochondrial transfer are 2 independent, but possibly interactive, pathways that allow their immunomodulatory effects [8,79]. Overall, this work indicates that NSC mitochondrial transfer is a novel strategy of cell-to-cell signalling that might support recovery in CNS disorders.

## Materials and methods

### Neural stem cells

Somatic NSCs were obtained from the subventricular zone (SVZ) of 7- to 12-week-old (18 to 20 g) C57BL/6 mice (Charles River, United Kingdom) and expanded in NSC media [Neuro-Cult basal medium (Stem Cell Technologies, Vancouver, Canada) plus mouse NeuroCult proliferation supplements (Stem Cell Technologies), 2 µg/ml of heparin (MilliporeSigma, Burlington, Massachusetts, United States of America), 20 ng/ml epidermal growth factor (EGF), and 10 ng/ml basic fibroblast growth factor (bFGF)], as described [10].

### Extracellular vesicle isolation

For EV isolation, NSCs were dissociated to single cells and plated at a concentration of $12 \times 10^6$ cells in 10 ml of EV medium in a T75 culture flask. EV medium was prepared with

DMEM/F12, glucose 30% w/v 21.92 ml/L, NaHCO$_3$ 7.5% w/v 16.44 ml/L, HEPES 1M 5.48 ml/L, Glutamax 1%, pen/strep 1%, EGF 20 ng/ml, FGF 10 ng/ml, heparin 0.2% w/v 2ml/L, apo-transferrin 96.14 mg/L, putrescine 3 mM 58μl/L, sodium selenite 3 mM 9.6μl/L, progesterone 2 mM 9.6μl/L, and insulin 24.16 mg/L (pre-dissolved in HCl 0.1 N), pH 7.4, as described [10,11]. After 18 hours, supernatants were collected and centrifuged for 15 minutes at 300 x $g$ to remove cellular pellets. The supernatant was collected and centrifuged for 15 minutes at 1,000 x $g$ to remove cellular debris. The supernatant was then collected and subjected to a first ultracentrifugation at 100,000 x $g$ for 70 minutes at 4˚C using an Optima XPN-80 ultracentrifuge with SW 32 Ti swinging rotor (Beckman Coulter, Brea, California, USA). Pellets were washed in PBS, and EVs were subjected to a final ultracentrifugation at 100,000 x $g$ for 30 minutes at 4˚C using an Optima MAX ultracentrifuge with a TLA-110 fixed angle rotor (Beckman Coulter).

## Isolation of CD63-enriched EV fractions

Further isolation of the CD63-enriched and depleted fractions from the EV preparation [10] was achieved using a commercially available magnetic-bead kit (Miltenyi Biotec, 130-117-041, Bergisch Gladbach, Germany) following the manufacturer's instructions. Briefly, after the first ultracentrifugation with the Optima XPN-80 ultracentrifuge, pelleted EVs were resuspended in 200 μl of PBS. The volume was then brought to 2 ml with PBS, and 50 μl of Exosome Isolation MicroBeads solution was added to the suspension. The EV-MicroBead suspension was incubated for 1 hour at room temperature (RT) on a shaking platform and then passed through a pre-wet μMACS Separator column (Miltenyi Biotec, 130-042-602) attached to a MACS MultiStand (Miltenyi Biotec, 130-042-303) following the manufacturer's instructions. The first eluent collected was designated as the CD63-depleted fraction (EVs$^{CD63\_depl.}$). The column was then removed from the magnetic MACS MultiStand, and the retained CD63-enriched fraction (EVs$^{CD63\_enrich.}$) was plunged into a clean microcentrifuge tube with 100 μl of isolation buffer. The resultant EVs$^{CD63\_depl.}$ and EVs$^{CD63\_enrich.}$ fractions were resuspended in a final volume of PBS (i.e., 3.2 ml) and then subjected to a final centrifugation at 100,000 x $g$ for 30 minutes at 4˚C using an Optima MAX ultracentrifuge with TLA-110 fixed angle rotor (Beckman Coulter). EVs$^{CD63\_depl.}$ and EVs$^{CD63\_enrich.}$ pellets were then resuspended in sterile PBS for downstream use.

## Exosome isolation

For exosome isolation, EV pellets (obtained as described above [10]) were resuspended in 0.5 ml 0.32 M sucrose. The solution was layered on a 10-ml continuous sucrose density gradient [0.32–2 M sucrose, 5 mM HEPES (pH 7.4)] and centrifuged for 18 hours at 100,000 x $g$ (SW 32.1 Ti; Beckman Coulter) with no brake. Fractions were collected from the top (low density) to the bottom of the tube (high density) of the sucrose gradient. Single fractions from the gradient were harvested, diluted in PBS, centrifuged at 100,000 x $g$ for 70 minutes at 4˚C using an Optima XPN-80 ultracentrifuge with SW 32 Ti swinging rotor (Beckman Coulter), and then processed for further analyses.

## Alternative EV isolation protocols

EV isolation with commercially available precipitation-based kits from Qiagen (cat. No. 76743, Hilden, Germany) and Thermo Fisher Scientific (cat. No. 4478359, Waltham, Massachusetts, USA) was carried out according to the manufacturers' instructions.

Briefly, for the Qiagen isolation kit, cell culture media was centrifuged at 3,000 x $g$ for 5 to 10 minutes to remove cells and debris. Cell culture supernatant was transferred to a fresh

microcentrifuge tube and mixed with Precipitation Buffer B for 60 minutes at 4˚C. Next, samples were centrifuged for 30 minutes at 20˚C, supernatant was removed, and the pellet was centrifuged again for 5 seconds. Finally, the pellet was resuspended in Resuspension Buffer and vortexed for 15 seconds. Samples were then processed for western blot (WB) analysis.

For the Invitrogen isolation kit, cell culture media was centrifuged at 2,000 x $g$ for 30 minutes to remove cells and debris. The supernatant was transferred to a new tube, and 0.5 volumes of the Total Exosome Isolation (from cell culture media) reagent was added. Then, the samples were incubated overnight at 4˚C. The samples were centrifuged at 10,000 x $g$ for 1 hour at 4˚C. Following centrifugation, the supernatant was removed, and the pellet resuspended for downstream WB analysis.

## Mitochondria depletion from EVs

The Wang and colleagues 2017 method, which includes a 0.22 μm filtration step (as previously described [25]), was used to remove most intact mitochondria from the EV preparation, which typically range between 0.2 and 1 μm. Briefly, supernatants from NSCs (cultured as above) were collected, transferred to 50 ml polypropylene centrifuge tubes, and centrifuged for 10 minutes at 300 x $g$. The supernatants were then collected and transferred into new 50-ml polypropylene centrifuge tubes before being subject to further centrifugation for 30 minutes at 2,000 x $g$. Subsequently, supernatants were transferred to 100-ml polycarbonate tubes and centrifuged for 20 minutes at 16,500 x $g$. Supernatants were transferred to new polycarbonate tubes prior to ultracentrifugation at 100,000 x $g$ for 70 minutes at 4˚C using an Optima XPN-80 ultracentrifuge with SW 32 Ti swinging rotor (Beckman Coulter). Pellets were immediately resuspended in ice-cold PBS and then filtered through a disposable filter unit (0.22 μm). Supernatants were pooled together in ultracentrifuge tubes and centrifuged for 30 minutes at 4˚C at 100,000 x $g$ using an Optima MAX ultracentrifuge with a TLA-110 fixed angle rotor (Beckman Coulter). Pellets resulting from this step, called EVs$^{Mito\text{-}depl.}$, were used for subsequent applications.

## Mitochondria-enriched preparation

Mitochondria-enriched preparations were obtained either from NSCs plated at $12 \times 10^6$ cells in 10 ml of complete growth media (CGM) (high density) or from NSCs plated at $1.5 \times 10^6$ cells in 10 ml of CFM (standard density), as previously described [20]. Briefly, cells were collected and centrifuged at 600 x $g$ for 8 minutes. The NSCs pellet was washed 2 times with ice-cold PBS and resuspended in 1 cell pellet volume of ice-cold 0.1X homogenisation medium IB (IB10X: 0.35M Tris–HCl, pH 7.8, 0.25 M NaCl, and 50 mM MgCl$_2$). Cells were then homogenised at 1,600 rpm for 3 minutes with a Teflon pestle, and 10X IB was immediately added at 1/10 of the initial cell pellet volume to maintain isotonic media. The homogenate was then centrifuged at 1,600 x $g$ for 3 minutes at 4˚C to pellet unbroken cells, debris, and nuclei. The supernatant was then collected and centrifuged at 13,000 rpm for 1 minute to obtain a mitochondrial enriched pellet. Once isolated, the mitochondrial pellet was washed with homogenisation medium 1X IB and centrifuged 1,300 rpm for 1 minute. Mitochondrial pellet was then resuspended in 200 μl of Medium A (0.32M sucrose, 1mM EDTA, and 10 mM Tris–HCl), centrifuged down, and resuspended in the appropriate volume of medium/PBS for downstream utilisation.

## Nanoparticle tracking analysis and tunable resistive pulse sensing analysis

EVs derived from $36 \times 10^6$ NSCs were diluted respectively 1:1,000 and 1:500 with PBS for NTA analysis using a Nanosight NS500 (Malvern Instruments) fitted with an Electron

Multiplication Charge-Coupled Device camera and a 532-nm laser. At least 3 videos were recorded for each sample using static mode (no flow). Between each capture, the sample was advanced manually, and the temperature was monitored and maintained at $25 \pm 1$°C. Data analysis was carried out on NTA 3.2 software using a detection threshold between 5 and 6. The concentration and size distribution of EVs was also analysed with TRPS (qNANO, Izon Science, Burnside, Christchurch, New Zealand) using a NP150 Nanopore at 0.5 V with 47-mm stretch. The concentration of particles was standardised using 100-nm calibration beads (CPC 100) at a concentration of $1 \times 10^{10}$ particles/ml. Data are presented as % of total particle per size distribution (nm) per ml.

## Nano flow cytometry analysis

After the first ultracentrifugation at 100,000 x $g$ for 70 minutes at 4°C using an Optima XPN-80 ultracentrifuge with SW 32 Ti swinging rotor (Beckman Coulter), EVs derived from $108 \times 10^6$ NSCs were resuspended in PBS and stained at 37°C for 30 minutes with MitoTracker Red CMXRos (Thermo Fisher Scientific, M7512, 300 nM) and Vio Bright FITC, REAfinity anti-mouse CD63 antibody (Miltenyi Biotec, 130-108-897, 1:10). Single stained and unstained controls were also made for later compensations. Paraformaldehyde (PFA) was then added to the samples to fix the EVs (3.7% final). Immediately following the addition of PFA, half of the suspension was passed through disposable filter unit (0.22 μm) to obtain EVs[Mito-depl.], while the other half was not (to obtain EVs). Samples were kept in PFA for 15 minutes at 37°C and then processed via a final centrifugation for 30 minutes at 4°C at 100,000 x $g$ using an Optima MAX ultracentrifuge with a TLA-110 fixed angle rotor (Beckman Coulter) as described above. The final pellets were resuspended in 25 μl of PBS and stored at 4°C until NanoFCM acquisition.

A NanoAnalyzer N30 instrument equipped with a dual 488/640 nm laser and single-photon counting avalanche photodiodes detectors (SPCM APDs) was used for detection of the EV isolates. Band-pass filters allowed for collection of light in specific channels (488/10 nm and 580/30 nm). HPLC grade water served as the sheath fluid via gravity feed, reducing the sample fluid diameter to approximately 1.4 μm. Data were generated through the NanoFCM Professional Suite v1.8 software, with noise being removed through the use of blanks. Measurements were taken over 1-minute periods at a sampling pressure of 1.0 kPa, modulated and maintained by an air-based pressure module. Samples were diluted in PBS as required to allow for 2,000 to 12,000 counts to be recorded during this time. During data acquisition, the sample stream is completely illuminated within the central region of the focused laser beam, resulting in approximately 100% detection efficiency, which leads to accurate particle concentration measurement via single-particle enumeration [26]. The concentration of samples was determined by comparison to 250-nm silica NP of known particle concentration to calibrate the sample flow rate. Isolated EV samples were sized according to standard operating procedures using the proprietary 4-modal silica nanosphere cocktail generated by nFCM to allow for a standard curve to be generated based on the 4 sizes of the nanosphere populations of 68 nm, 91 nm, 113 nm, and 155 nm in diameter. Silica provides a stable and monodisperse standard with a refractive index of approximately 1.43 to 1.46, which is close to the range of refractive indices reported in the literature for EVs ($n$ = 1.37 to 1.42) [26]. Using such a calibration standard enables accurate flow cytometry size measurements, as confirmed when comparing flow cytometry with cryo-TEM results [27]. The laser was set to 10 mW and 10% SSC decay. Data reported in the figure were handled within the nFCM Professional Suite v1.8 software and FlowJo to analyse particles between 40 nm and 1,000 nm and gate for the proportional analysis of subpopulations.

## Proteomic analysis sample preparation

EVs and exosomes were purified from murine NSC culture supernatants by ultracentrifugation (total particles) followed by density gradient centrifugation (exosomes) as previously described [10]. For comparison with NSCs, whole-cell lysates (NSCs) and washed NSC cell pellets were processed in parallel. All samples were prepared in triplicate (biological replicates). Tryptic digests were made using an IST-NHS sample preparation kit (Preomics GmBH, Germany) according to the manufacturer's instructions with minor modifications. Briefly, samples were solubilised in proprietary lysis buffer and sonicated 10 times (30-second on/off) in a Bioruptor sonicator (Diagenode, Denville, New Jersey, USA). Lysates were diluted 10-fold and quantified by BCA protein assay against a bovine serum albumin (BSA) standard curve in diluted lysis buffer. Digestion was performed at 37°C for 3 hours. TMT labelling was performed on the digestion columns as per the manufacturer's instructions. After elution, TMT labelling of at least 98% peptides was confirmed for each sample before pooling and subjecting to high pH reversed phase (HpRP) fractionation. This was conducted on an Ultimate 3000 UHPLC system (Thermo Fisher Scientific) equipped with a 2.1 mm × 15 cm, 1.7μ Acquity BEH C18 column (Waters, UK). Solvent A was 3% acetonitrile (ACN), solvent B was 100% ACN, and solvent C was 200 mM ammonium formate (pH 10). Throughout the analysis, solvent C was kept at a constant 10%. The flow rate was 400 μl/min, and UV were monitored at 280 nm. Samples were loaded in 90% A for 10 minutes before a gradient elution of 0% to 10% B over 10 minutes (curve 3), 10% to 34% B over 21 minutes (curve 5), and 34% to 50% B over 5 minutes (curve 5) followed by a 10-minute wash with 90% B. Moreover, 15-second (100 μl) fractions were collected from the start of the gradient elution. Fractions were pooled orthogonally to generate a final 24 fractions.

## Mass spectrometry

All samples were resuspended in 5% dimethyl sulfoxide/0.5% trifluoroacetic acid. Samples were analysed using a nanoLC-MS platform consisting of an Ultimate 3000 RSLC nano UHPLC (Thermo Fisher Scientific) coupled to an Orbitrap Fusion (Thermo Fisher Scientific) instrument. Samples were loaded at 10 μl/min for 5 minutes onto an Acclaim PepMap C18 cartridge trap column (300 μm x 5 mm, 5 μm particle size) in 0.1% TFA. After loading, a linear gradient of 3% to 32% solvent B over 180 minutes was used for sample separation over a column of the same stationary phase (75 μm x 75 cm, 2 μm particle size) before washing at 95% B and equilibration. Solvents were A: 0.1% formic acid (FA) and B: 100% ACN/0.1% FA. Electrospray ionisation was achieved by applying 2.1kV directly to a stainless-steel emitter tip. Instrument settings were as follows. MS1: Quadrupole isolation, 120'000 Resolution, 5e5 AGC target, 50-ms maximum injection time, ions injected for all parallisable time. MS2: Quadrupole isolation at an isolation width of m/z 0.7, CID fragmentation (NCE 30) with ion trap scanning out in rapid mode from m/z 120, 5e3 AGC target, 70-ms maximum injection time, ions accumulated for all parallisable time in centroid mode. MS3: In Synchronous precursor selection mode, the top 10 MS2 ions were selected for HCD fragmentation (65NCE) and scanned out in the orbitrap at 50'000 resolution with an AGC target of 2e4 and a maximum accumulation time of 120 ms. ons were not accumulated for all parallelisable time. For all experiments, the entire MSn cycle had a target time of 3 seconds.

Spectra were searched by Mascot within Proteome Discoverer 2.2 in 2 rounds of searching. The first search was against the UniProt Mouse reference proteome and a compendium of common contaminants (Global Proteome Machine Organization). The second search took all unmatched spectra from the first search and searched against the Mouse trEMBL database. Search parameters were as follows. MS1 Tol: 10 ppm, MS2 Tol: 0.6 Da, Fixed mods:

Carbamidomethyl (C) and TMT (N-term, K), Var mods: Oxidation (M), Enzyme: Trypsin (/P). For HCD-OT Experiments. MS1 Tol: 10 ppm, MS2 Tol: 0.05 Da, Fixed mods: Carbamidomethyl (C) and TMT (N-term, K), Var mods: Oxidation (M), Enzyme: Trypsin (/P). MS3 spectra were used for reporter ion-based quantitation with a most confident centroid tolerance of 20 ppm. PSM false discovery rate (FDR) was calculated using Mascot percolator and was controlled at 0.01% for "high" confidence PSMs and 0.05% for "medium" confidence PSMs. Normalisation was automated and based on total s/n in each channel. Proteins/peptides satisfying at least a "medium" FDR confidence were taken forth for further analysis. To compare protein abundances in particles, exosomes, and NSCs, moderated *t* tests were performed using the *limma* R/Bioconductor software package, with FDR-adjusted *p*-values (q values) calculated according to the Benjamini–Hochberg method. To analyse subcellular localisations of proteins identified or enriched in particles and/or exosomes, GOCC terms were imported using the Perseus software platform. Further data manipulation and general statistical analysis were conducted using Excel and XLSTAT. The proteomic data described in this study have been deposited to the ProteomeXchange consortium via the PRIDE partner repository (accessible at http://proteomecentral.proteomexchange.org), with the dataset identifier PXD024368.

## Western blotting

Whole-cell pellets were solubilised in 100 μl of RIPA buffer (10 mM Tris HCl pH 7.2, 1% v/v sodium deoxycholate, 1% v/v Triton X-100, 0.1% v/v sodium dodecyl sulfate (SDS), 150 mM NaCl, 1 mM EDTA pH 8) in the presence of Complete Protease Inhibitor Cocktail (Roche, Basel, Switzerland) and Halt Phosphatase Inhibitor Cocktail (Thermo Fisher Scientific). EV, exosome, and isolated mitochondria pellets were solubilised in 50 μl of RIPA buffer with 3% v/v SDS. Protein abundance was quantified using the Bio-Rad DC Protein Assay Kit II (Bio-Rad, Hercules, California, USA).

Protein extracts were then mixed with NuPAGE 4x LDS sample buffer (Invitrogen) under reducing (10× sample reducing agent) and nonreducing conditions and separated by SDS-PAGE on 4% to 12% precast NuPAGE Bis/Tris gels (1.5-mm thickness) or on 4% to 15% precast mini-PROTEAN TGX stain-free gels. Before loading, samples were heated at 95˚C for 5 minutes, then loaded onto the gels, and then run at 120V in either MOPS or 10x Tris-Glycine SDS (Bio-Rad, 161-0732EDU) running buffer. Samples were then transferred on polyvinylidene fluoride membrane (0.45 μm pore size, Immobilon) filter paper sandwich using XCell II Blot Module and NuPAGE transfer buffer (Invitrogen) or a Trans-Blot Turbo Transfer System (Bio-Rad).

For immunoblot analysis, the membranes were blocked with 5% nonfat milk in 0.1% PBS-Tween 20 (MilliporeSigma) for 1 hour at RT and then incubated with the following primary antibodies (diluted in 5% nonfat milk in 0.1% PBS-Tween 20) for 18 hours at 4˚C: Mouse monoclonal Total OXPHOS rodent WB cocktail (Abcam, Cambridge, UK, ab110413, 1:1,000); mouse monoclonal Total OXPHOS blue-native WB cocktail (Abcam, ab110412, 1:1,000); rat monoclonal anti-CD9 (BD Biosciences, Franklin Lakes, New Jersey, USA, 553758, 1:1,000); anti-CD63 (MBL International, Woburn, Massachusetts, USA, D263-3, 1:500); mouse monoclonal anti-Pdcd6ip (AIP-1/Alix) (BD Biosciences, 611620, 1:500); goat polyclonal anti-TSG101 (Santa Cruz, Dallas, Texas, USA, sc-6037, 1:500); rabbit monoclonal TOMM20 (Santa Cruz, sc-11415,1:1,000); rabbit polyclonal ant-H3 (Abcam, ab1791,1:10,000), mouse monoclonal anti-β-actin (MilliporeSigma, A1978, 1:10,000), and mouse monoclonal anti-Gm130/Golga2 (BD Biosciences, 610823, 1:1,000). Molecular weight marker: SeeBlue Plus2 (Invitrogen, LC5925).

After primary antibody incubation, membranes were washed 3 times for 10 minutes with 0.1% PBS-Tween 20 and incubated with the appropriate horseradish-peroxidase-conjugated

secondary antibodies (Thermo Fisher Scientific and Cell Signaling Technology, Danvers, Massachusetts, USA) for 1 hour at RT. If necessary, the membranes were then subjected to a stripping protocol. Briefly, after image acquisition, the membrane was washed with PBS and then incubated for 1 hour with stripping buffer (10x: 75.08 g Glycine [MilliporeSigma, 410225], 10 g SDS [MilliporeSigma, 862010], 1L dH$_2$O, pH 7.4). After 2 washes with 0.1% PBS-Tween 20, the membrane was then blocked with 5% nonfat milk in 0.1% PBS-Tween 20 plus for 1 hour and then incubated with the next relevant primary antibodies (diluted in 5% nonfat milk in 0.1% PBS-Tween 20). The secondary antibodies used for all the WBs were goat anti rabbit-HRP (Thermo Fisher Scientific, 31460, 1:10,000 and Cell Signaling Technology, 7074, 1:2,000), goat anti mouse-HRP (Thermo Fisher Scientific, 31430, 1:20,000 and Cell Signaling Technology, 7076, 1:2,000), rabbit anti-goat (Thermo Fisher Scientific, 31402, 1:10,000), and goat anti-rat (Thermo Fisher Scientific, 31470, 1:10,000).

Immunoreactivity was revealed by using Western Lightning Plus-ECL Prime Western Blotting Detection Reagent (General Electric Healthcare,Chicago, Illinois, USA, #RPN2232) or Clarity Western ECL Substrate (Bio-Rad, 1705061) according to the manufacturer's instruction. Images were acquired on a Bio-Rad Chemidoc MP system using high sensitivity acquisition in signal accumulation mode.

## Genomic PCR

Total DNA was extracted from cell pellets derived from $3 \times 10^6$ NSCs and from L929 Rho$^0$, EVs derived from $12 \times 10^6$ NSCs, and isolated mitochondria derived from $12 \times 10^6$ NSCs. Prior to DNA isolation, samples were resuspended in PBS and suspensions pretreated with DNase I (MilliporeSigma, #D5025) [50 U/ml] for 1 hour at 37˚C to remove possible external contaminating DNA. Nuclease reaction was stopped by a PBS wash at the appropriate centrifugation speed for each sample (isolated mitochondria preparation: 12,400 x $g$ for 1 minute; NSCs: 16,000 x $g$ for 5 minutes; EVs: 100,000 x $g$ for 45 minutes [ultracentrifugation steps were performed with a TLA-110 rotor (Beckman Coulter)]). DNase I pretreated samples were all resuspended in 200 μl of PBS, and total DNA was finally extracted with DNeasy Blood & Tissue Kit (Qiagen) according to manufacturer's instructions. Total DNA yield and purity of the extracted DNA were determined using a Nanodrop spectrophotometer (Thermo Fisher Scientific).

The nuclear and mitochondrial DNA (nDNA and mtDNA) content of each sample was assessed by PCR amplification of the mtDNA gene *mt-ND1* and the nuclear DNA gene *Sdhd*, starting from 25 ng of total DNA extract and using KAPA biosystem 2G kit according to manufacturer's instructions. Primer sequences: *mt-ND1*—F: 5′-TGCACCTACCCTATCACTCA-3′; R: GGCTCATCCTGATCATAGAATGG; expected product: 148 bp; *Sdhd*—F: CTTGAA TCCCTGCTCTGTGG; R: AAAGCTGAGAGTGCCAAGAG; expected product: 1660 bp. Amplification reaction parameters: initial denaturation at 95˚C for 3 minutes (1x); denaturation 95˚C for 15 seconds, annealing temperature 60˚C for 15 seconds, elongation time (ET) 72˚C for 1 second (all 35x); final extension 72˚C for 1 minute. Amplified nucleic acids (25 μl) were mixed with 6X Purple dye loading buffer (BioLegend, USA) loaded onto a 2% Agorose/Gel Red (BioLegend,) /1x TAE gel and run for 40 minutes at 120V. Loading Buffer: TAE 1X electrophoresis buffer. At run completion, DNA bands were visualised with UV light (312 nm) with Bio-Rad ChemiDoc system and directly photographed.

## Transmission electron microscopy

For TEM analysis, EVs were generated and isolated from NSCs following the protocol described above [10]. Following the last ultracentrifugation step, the supernatant was carefully removed, and the pellets were fixed with 4% PFA-PBS for 10 minutes at RT, and then kept

overnight at 4˚C. After removal of the fixative and a short rinse with PBS, the pellets were post-fixed in 1% $OsO_4$ (Taab, Aldermaston, Berks, UK) for 30 minutes, rinsed with distilled water, dehydrated in graded ethanol, including block staining with 1% uranyl acetate in 50% ethanol for 30 minutes, and embedded in Taab 812 (Taab). Overnight polymerisation at 60˚C was followed by sectioning, and the ultrathin sections were analysed using a Hitachi 7100 electron microscope (Hitachi, Chiyoda City, Tokyo, Japan) equipped with Veleta, a $2,000 \times 2,000$ MegaPixel side mounted TEM CCD camera (Olympus, Shinjuku City, Tokyo, Japan).

## Cryo-transmission electron microscopy

For immuno-gold labelling and cryo-TEM analysis of EVs, 2 independent methods were used (i.e., with and without permeabilisation).

In the first method, 10-nm gold NP were conjugated with anti-TOMM20 monoclonal antibody (Abcam, ab232589) following the procedures previously described [28]. Following the last ultracentrifugation step, resuspended EV pellets were labelled for 1 hour with anti-TOMM20-gold-NP at 1 to $3 \times 10^{15}$ gold-NP/L. Immuno-gold labelled EV samples were processed for cryo-TEM according to standard procedures. Briefly, a 4-μl aliquot of EVs was deposited on an EM grid coated with a perforated carbon film. After draining the excess liquid with a filter paper, grids were plunge-frozen into liquid ethane cooled by liquid nitrogen using a Leica EMCPC cryo-chamber (Leica Microsystems, Wetzlar, Germany). For cryo-TEM observation, grids were mounted onto a Gatan 626 cryoholder and transferred to a Tecnai F20 microscope (Thermo Fisher Scientific) operated at 200 kV. Images were recorded with an Eagle 2k CCD camera (FEI, USA).

In the second method, 10-nm gold NPs were conjugated with anti-CD63 (MBL International, D263-3) and 20-nm gold NPs were conjugated with anti-TOMM20 (Abcam, ab232589) using commercially available gold conjugation kits (Abcam, ab201808 [10 nm] and ab188215 [20 nm]). Following the first 100,000 x $g$ ultracentrifugation step, EV pellets were resuspended and incubated in a solution of PBS and 0.01% saponin (Alfa Aesar, Haverhill, Massachusetts, USA, J63209, 1:50), as described [80]. Anti-CD63 and anti-TOMM20 gold-conjugated NPs were added at 1:10 concentration, and samples were put on a benchtop shaker set to 700 rpm for 1 hour at RT. EVs were then subjected to a final centrifugation at 100,000 x $g$ for 30 minutes at 4˚C using an Optima MAX ultracentrifuge with a TLA-110 fixed angle rotor (Beckman Coulter). The final EV pellet was resuspended in 15 μl PBS, and immuno-gold labelled EV samples were processed for cryo-TEM according to standard procedures without thin sectioning. Briefly, EVs were prepared through vitrification by plunge freezing of the aqueous suspensions on copper grids (300 mesh) with lacey carbon film. Prior to use, the grids were glow discharged using a Quorum Technologies GloQube instrument (Quorum Technologies, East Sussex, UK). Suspensions of the samples (2.5 μl) were put onto the grid, blotted using dedicated filter paper, and immediately frozen by plunging in liquid ethane utilising a fully automated and environmentally controlled blotting device, Vitrobot Mark IV. The Vitrobot chamber was set to 4˚C and 95% humidity. Specimens after vitrification were kept under liquid nitrogen until they were inserted into a Gatan Elsa cryo holder and analysed in the TEM at −178˚C. Cryo-TEM micrographs were obtained using a Thermo Fisher Scientific (FEI) Talos F200X G2 microscope equipped with a field emission gun operating at an acceleration voltage of 200 kV. Images were recorded on a Ceta 4k x 4k CMOS camera and processed with Velox software.

## Blue native polyacrylamide gel electrophoresis

Extractions of proteins in native conditions from EVs, isolated mitochondria, and NSCs were performed as previously described [30].

## EV native protein extraction

EVs derived from $36 \times 10^6$ NSCs were incubated on ice for 10 minutes in hypotonic buffer solution (EDTA-Na 1 mM, Tris HCl 6 mM, pH 8) with added Protease Inhibitor Cocktail 1X. Next, EVs were homogenised with a pestle and isotonic conditions were restored by adding sorbitol 1 M (final concentration 0.32 M), EDTA 0.5 M (final concentration 10 mM), and Tris-HCl 1 M (final concentration 10 mM). The extract was then centrifuged at 10,000 x $g$ for 10 minutes, washed with PBS, and centrifuged again at 10,000 x $g$ for 5 minutes. The pellet was resuspended with Medium A supplemented with BSA and centrifuged 10,000 x $g$ for 10 minutes. This pellet was then resuspended in 50 μl of Medium A, and 5 μl was taken for BCA protein quantification. Ultimately, the extracts were centrifuged again at 10,000 x $g$ for 10 minutes and resuspended at a protein concentration of 5 mg/ml in solubilisation buffer (1.5 M aminocaproic acid, 50 mM Bis-Tris/HCl pH = 7.0). For the solubilisation, 1.6 mg of n-dodecyl-β-D-maltoside (DDM) per mg of protein was added. Samples were incubated on ice for 5 minutes, then centrifuged at 20,000 x $g$ for 30 minutes at 4°C. Supernatants were collected, and a same volume of sample buffer (750 mM aminocaproic acid, 50 mM Bis–Tris/HCl, pH 7.0, 0.5 mM EDTA, and 5% Serva Blue G) was added. The samples were stored at −80°C until it was time to perform BN-PAGE.

## NSC native protein extractions

Pellets from $3 \times 10^6$ cells were resuspended in 200 μl of PBS. A total of 200 μl of cold digitonin solution (8 mg/mL dissolved in PBS) was added to the sample and kept on ice. After 10 minutes of incubation, the digitonin solution was diluted by adding 1 ml of cold PBS, and cells were centrifuged at 10,000 x $g$ for 5 minutes at 4°C. This step was repeated twice. Cell pellets were then resuspended in 100 μl of PBS, and 5 μl was taken for protein quantification. Next, PBS was removed by centrifuging at 10,000 x $g$ for 5 minutes, and the cell pellet was solubilised as described for EV samples.

## Isolated mitochondria native protein extraction

The mitochondrial pellet was resuspended in 100 μl of PBS, and 5 μl was taken for protein quantification. Next, Medium A was removed by centrifuging at 10,000 x $g$ for 5 minutes, and the pellet containing the isolated mitochondria was solubilised and processed as previously described for EVs and NSCs.

A total of 50 μg of protein for each of the samples was loaded into a pre-cast NativePAGE 3–12% Bis-Tris gel (Invitrogen). For the run, 1x NativePAGE Running Buffer plus 1x Native-PAGE Cathode Buffer Additive (Invitrogen) was added to the cathode, and 1x NativePAGE Running Buffer was added to the anode. Halfway through the run, the cathode buffer was substituted for 1x NativePAGE Running Buffer plus 0.1x NativePAGE Cathode Buffer Additive and run until the front reached the end of the gel.

## In-gel mitochondrial complexes activity

After the electrophoresis, gels were incubated with the following buffers containing the substrates and electron acceptors necessary for the colorimetric reactions to take place [31]. Complex I: 1 mg/ml nitro blue tetrazolium (NBT), 1 mg/ml NADH in 5 mM Tris-HCl, pH 7.4. Complex II: 1 mg/ml NBT, 20 mM sodium succinate, 0.2 mM phenazine methasulfate (PMS) in 5 mM Tris-HCl, pH 7.4. Complex IV: 1 mg/ml 3,3′-diaminobenzidine tetrahydrochloride (DAB), 24 U/ml catalase, 1 mg/ml cytochrome c, 75 mg/ml sucrose in 50 mM potassium phosphate buffer, pH 7.4.

## Mitochondrial membrane potential measurement

Mitochondrial membrane polarisation was determined using MitoProbeTM JC-1 Assay Kit for Flow Cytometry (Thermo Fisher Scientific, M34152) according to the manufacturer's instructions. For JC-1 (5,5,6,6′-tetrachloro-1,1′,3,3′-tetraethylbenzimidazolylcarbocyanine iodide) staining, $6 \times 10^6$ NSCs, as well as EVs and mitochondria derived from $6 \times 10^6$ NSCs, were resuspended in 500 µl of NSC medium and loaded with JC-1 (2 µM) and incubated for 30 minutes at 37˚C (5% $CO_2$). After incubation, samples were washed once, resuspended in 400 µl PBS, plated in quadruplicate (100 µl/1M/well) in a 96-well flat bottom plate, and immediately analysed on a Tecan Infinite M200 Pro plate reader. J-aggregated red fluorescence was read at Ex488/Em590, while monomers with green fluorescence were read at Ex488/Em530. A Blank signal from PBS only was subtracted from sample fluorescence values, and data were expressed as mean JC1 ratio Red-over-Green fluorescence (JC1-R/G) (± SEM) per unit (unit defined as $10^6$ NSCs and the EVs/Mito produced by $10^6$ NSCs).

## High-resolution respirometry (Oroboros oxygraph respiration assay)

Mitochondrial respiration in permeabilised NSCs and hypotonic shock-treated EVs was analysed by high-resolution respirometry (HRR) (Oroboros Instruments, Innsbruck, Austria). HRR analysis was performed based on previously published methods [29,81]. Briefly, a 2 ml oxygraph (Oxygraph-2k; Oroboros Instruments) chamber was washed with 70% ethanol, rinsed 3 times with distilled water, then filled with the respiration medium (Medium A plus 1 mM adenosine diphosphate (ADP), 2 mM potassium phosphate, and 1 mg/ml fatty acid free BSA) to be used in each of the assays. Moreover, $20 \times 10^6$ NSCs were resuspended in 2 ml of Medium A (20 mM HEPES (adjusted to pH 7.1 with NaOH or KOH), 250 mM sucrose, 10 mM $MgCl_2$). For permeabilisation, the cell suspension was incubated with 5 µl of 1% digitonin solution for 1 minute at RT on a tube oscillator. Digitonin was then diluted with 5 mL of Medium A and removed by centrifugation at 1,000 x $g$ for 3 minutes. Finally, the pellet was resuspended in 2.1 ml of respiration medium and added into the chamber.

EVs from $36 \times 10^6$ NSCs were isolated and subjected to hypo-osmotic shock as described above. Upon removal of homogenisation Medium A through centrifugation at 10,000 x $g$ for 10 minutes, hypotonic-shock treated vesicles were equilibrated in MAITE medium (25 mM sucrose, 75 mM sorbitol, 100 mM KCl, 0.05 mM EDTA, 5 mM $MgCl_2$, 10 mM Tris-HCl, 10 mM phosphate, pH 7.4). The suspension was centrifuged at 10,000 x $g$ for 10 minutes, resuspended in 100 µl of MAITE buffer, then diluted up to 2.1 mL with MAITE buffer plus 1 mg/ml fatty acid free BSA.

Once samples were loaded in the chambers, a polyvinylidene fluoride stopper was inserted to generate a closed system with a final volume of 2 mL. Oxygen concentration was recorded at 0.5 Hz and converted from voltage to oxygen concentration using a 2-point calibration. Respiration rates ($O_2$ flux) were calculated as the negative time derivative of oxygen concentration (Datlab Version 4.2.1.50, Oroboros Instruments). The $O_2$ flux values were corrected for the small amount of back diffusion of oxygen from materials within the chamber, any leak of oxygen from outside of the vessel, and oxygen consumed by the polarographic electrode. A protocol involving serial additions of selected mitochondrial complex substrates and inhibitors was performed for a comprehensive assessment of mitochondrial function. The following substrates and inhibitors were progressively injected with a Hamilton syringe at the respective concentrations: 5 mM glutamate and 5 mM malate (NADH-linked substrates); 0.1 µM rotenone (complex I inhibitor); 5 mM glycerol-3-phosphate and 5 mM succinate (FADH2-linked substrate); cytochrome $c$ (Cyt c) to compensate for a possible loss due to outer membrane disruption; 20 nM antimycin A (complex III inhibitor); 1 mM $N,N,N′,N′$-tetramethyl-$p$-

phenylenediamine (TMPD, complex IV electron donor) and finally 0.1 mM KCN (complex IV inhibitor). Data are shown as OCR $pmolO_2$ normalised by the EVs produced by $10^6$ NSCs.

## Lentiviral particles generation and NSCs transduction

To label NSCs, cells were transduced in vitro using a third-generation lentiviral carrier (pRRLsinPPT-hCMV) coding for the enhanced farnesylated (f) green fluorescent protein (GFP), which targets the FP to the inner plasma membrane of transduced cells [82]. To label the mitochondria of NSCs, cells were transduced in vitro using a third generation lentiviral (pLL3.7) carrier coding for enhanced MitoDsRed expression (i.e., a fusion protein that encodes the leader sequence of cytochrome oxidase IV linked to the florescent protein DsRed), which targets the FP to the mitochondrial matrix [32–34]. The functional stability of these cells in the absence or in the presence of the lentiviral transcript has been confirmed with clonal and population studies [48]. Briefly, neurospheres were harvested, dissociated to a single cell suspension, and seeded at a high density [$1.5 \times 10^6$ in a T75 flask (MilliporeSigma)] in 5-ml fresh medium. After 12 hours, $3 \times 10^6$ T.U./ml of lentiviral vectors were added, and 6 hours, later additional 5 ml of fresh medium were added. Seventy-two hours after viral transduction, cells were harvested, reseeded at normal concentration, and transgene expression was measured by flow cytometric analysis.

## L929-Rho$^0$ experiments

Mouse L929-Rho$^0$ fibroblasts were grown as adherent cells in fibroblasts medium [DMEM, high glucose, GlutaMAX Supplement (Thermo Fisher Scientific), pyruvate (Thermo Fisher Scientific), 10% fetal bovine serum (FBS), 1% pen/strep (Invitrogen) and uridine (Millipore-Sigma) 50 μg/ml] until they reached confluency (80% to 90%). The day of passage, cells were washed with PBS (Gibco). Trypsin (0.05% in DMEM) was added at 37˚C and inactivated after 3 minutes with fibroblast medium (2:1). Cells were collected and spun at 200 x $g$ for 5 minutes and then reseeded 1:5 in T175 flasks for normal expansion.

L929-Rho$^0$ cells were collected, centrifuged at 200 x $g$ for 5 minutes, then seeded on 13-mm cover slips ($5 \times 10^3$ cells/coverslips, BD Biosciences) in 24-well plates in 200 μl of uridine supplemented fibroblast medium. The day after, culture media was removed and substituted with either selective medium (Uridine$^-$) or complete media according to experimental design. Media changes with selective or complete fibroblast medium was carried out every 24 hours to remove dead cells and/or to supplement with fresh uridine where needed. Three days following the switch to selective media, phenotype rescue experiments were carried out by adding EVs or mitochondria obtained from MitoDsRed$^+$ NSCs resuspended in selective medium (ratios = 1 Rho$^0$ L929 fibroblast: EVs/Mito collected from 30 or 90 MitoDsRed$^+$ NSCs). Medium was removed and substituted with selective medium (Uridine$^-$) or with complete media every 24 hours until the end of experiment.

We carried out fluorescence stainings and quantification of uptake of MitoDsRed$^+$ particles (3 hours after treatment) and Rho$^0$ L929 survival (5 days after treatment). Rho$^0$ L929 fibroblasts were rinsed with PBS and then permeabilised with PBS plus 0.1% Triton X-100. To selectively label F-actin, cells were stained with Alexa Fluor 488 phalloidin (Thermo Fisher Scientific, #A12379, 1:200) in 0.1% Triton X-100 for 20 minutes at RT, and then washed with PBS. Nuclei were counterstained with 4′,6-diamidino-2-phenylindole (DAPI) (1:10,000, Invitrogen) for 3 minutes, and coverslips were mounted with Dako mounting kit (Agilent Technologies, Santa Clara, California, USA). Uptake of MitoDsRed$^+$ particles was quantified with confocal microscopy by counting the number of cells with MitoDsRed$^+$ inclusions over the

total number of DAPI$^+$ cells using a 20× objective on 6 region of interests (ROIs) per $N = 4$ replicates per condition.

Rho$^0$ L929 survival was quantified using 4× objective images of the entire coverslip area captured on an Olympus BX53 microscope with motorised stage and Neurolucida software. Images were analysed using ImageJ software. Data were represented as number of total DAPI$^+$ cells/area, $N = 4$ replicates per condition.

For Sanger sequencing, the DNA from NSCs and Rho$^0$ fibroblast L929 cells was extracted at 16 days after treatment, and the mitochondrial encoded gene *ND3* (*mt-ND3*) was PCR-amplified using KAPA polymerase kit (MilliporeSigma) following manufacturer's instruction. The primers used were mt-ND3 forward, 5′-TTCCAATTAGTAGATTCTGAATAAACCCA GAAGAGAGTAAT-3′ and mt-ND3 reverse 5′-CGTCTACCATTCTCAATAAAATTT-3′. The PCR amplification protocol is described above. All PCR products were first evaluated on 2% agarose gels (as described above) and then purified using the QIAquick GelExtraction Kit (Qiagen). Sanger sequencing of these products was performed with reverse primers by Source Bioscience (www.sourcebioscience.com). Obtained sequences were visualised and analysed with 4peaks (https://nucleobytes.com/4peaks/index.html).

## Bone marrow–derived macrophages culture

Bone marrow–derived Mφ were obtained from the bone marrow of C57BL/6 mice, as previously described [37].

## Analysis of endocytosis/phagocytosis

Mφ were plated in 12-well plates at 150,000 cells per well in 1 ml of Mφ medium (DMEM high glucose + 10% dialysed FBS + 1% pen/strep) + 10% mCSF. Cells were let to rest overnight and stimulated with 50 ng/ml LPS (Enzo Life Sciences, Farmingdale, New York, USA). After 30 minutes from the start of the stimulation, cells were treated with either 50 μM dynasore, 20 μM Pitstop 2 (Abcam), or 10 μM cytochalasin D (Thermo Fisher Scientific). Moreover, 30 minutes after drug treatment, the cells received either EVs that were extracted from MitoDsRed$^+$ NSCs at a ratio of 1 Mφ: EVs collected from 30 MitoDsRed$^+$ NSCs. Fluorescent labelled 0.04 μm latex beads-Alexa 647 (Thermo Fisher Scientific) at a concentration of 1:100,000, or fluorescent labelled liposomes-Alexa 647 (MilliporeSigma) at a concentration of 0.8 μM L-a-phosphatidylcholine, 0.2 μM stearylamine, and 0.1 μM cholesterol per well were used as controls. After 15 minutes of treatment, the wells were washed with PBS, fixed with 4% PFA-PBS, and stained with a primary rat anti-F4/80 antibody (Bio-Rad, 1:100) followed by an anti-rat secondary antibody and DAPI (1:10,000), before being mounted. Images were taken with a Dragonfly Spinning Disk imaging system (Andor Technologies, Belfast, Northern Ireland, UK) composed by a Nikon Ti-E microscope, Nikon 60x TIRF ApoPlan, and a Zyla sCMOS camera, and analysed using the Imaris v.9.1.2 software (Bitplane AG, Zurich, Switzerland). Internalised EVs, latex beads, and liposomes were counted from $N \geq 10$ cells per $N \geq 8$ ROIs per condition.

## Fluorescent-activated cell sorting analysis

Exogenous mitochondria uptake was assessed in LPS-stimulated Mφ at 6 hours following MitoDsRed$^+$ EV or MitoDsRed$^+$ exosome treatment. Mφ were plated in 6-well plates (500,000 cells/well), left to rest overnight, and stimulated with 50 ng/ml LPS (Enzo Life Sciences). After 1 hour of LPS stimulation, EVs and exosomes isolated from MitoDsRed$^+$ NSCs were resuspended in Mφ medium and added to the same well (ratios: 1 Mφ: EVs collected from 30 MitoDsRed$^+$ NSCs). After 6 hours, 500 μl of Accumax (MilliporeSigma, A7098) was added to

each well, and cells were detached and collected by gentle scraping followed by a single wash with 500 μl of PBS of the well. Cells were then centrifuged at 1,300 x *g* for 5 minutes, stained with an anti-F4/80 APC/Cy7 antibody (BioLegend, 123117), resuspended in 200 μl of PBS, and transferred into FACS tube for analysis. Subsequently, cells were analysed by flow cytometry using a BD LSR Fortessa (BD Biosciences) operated by BD FACS Diva software, and at least 100,000 events were collected per sample.

## Quantification of mitochondria from EVs included in lysosomes, peroxisomes, and mitochondrial network

To analyse the integration of exogenous mitochondria, Mφ were plated in an 8-well chamber slide (Nunc Lab-Tek II Chamber Slide System, Thermo Fisher Scientific) (50,000 cells/well) and left to rest overnight.

For lysosome and peroxisome staining, Mφ were treated as above with LPS and EVs (isolated from MitoDsRed[+] NSCs). After 6 hours of incubation, cells were washed with PBS 3 times and fixed with 4% PFA. Anti-LAMP1 (Abcam ab24170,1:500) or PMP70 (Sigma, SAB420018, 1:500) primary antibodies were used to stain for lysosomes or peroxisomes, respectively. Images were acquired using a Dragonfly Spinning Disk system as described above but using a Zyla sCMOS camera instead. MitoDsRed[+] EV co-localisation analysis was performed in $N \geq 10$ cells per condition using Imaris Software, and X coefficient was calculated ($N = 2$ independent experiments).

For mitochondria staining, Mφ were exposed to 100 nM of MitoTracker Green FM (Thermo Fisher Scientific, M-7514) for 10 minutes and then washed with media. Mφ were then stimulated with 50 ng/ml LPS (Enzo Life Sciences), and after 1 hour, EVs isolated from MitoDsRed[+] NSCs were resuspended in Mφ medium and added in the same well (ratios: 1 Mφ: EVs collected from 30 MitoDsRed[+] NSCs). At 6 hours of incubation, live images and videos (0.15 frames per second, fps) were acquired using a Dragonfly Spinning Disk imaging system (Andor Technologies) composed by a Nikon Ti-E microscope, Nikon 60x TIRF ApoPlan, and a Ixon EMCCD camera. The videos were processed (8 FPS), and the images analysed using the Imaris v.9.1.2 software (Bitplane AG, Zurich, Switzerland). The 3D surface and spots were created for the mitochondrial network and the MitoDsRed[+] EV, respectively. The distance between the surface (mitochondrial network) and spots (MitoDsRed[+] EV) were calculated using 2 different thresholds ($\leq 0$ μm to count for inclusion of EVs into the network and $>0$ μm,$<0.5$ μm for attaching). The analysis of $N \geq 5$ cells per condition was done blinded ($N = 3$ independent experiments).

## Split self-associating fluorescent protein experiments

Plasmids pSFFV_sfCherry2(1–10) (Addgene plasmid #82603; http://n2t.net/addgene:82603; RRID:Addgene_82603) and pEGFP_TOMM20_sfCherry2(11) (Addgene plasmid #83033; http://n2t.net/addgene:83033; RRID:Addgene_83033), expressing components of the sfCherry2 split FP system were used [38]. Agar stabs containing *Escherichia coli* transformed with the plasmids of interest were grown on LB agar plates overnight. Individual colonies were picked and expanded in LB broth overnight. Plasmid DNA from *E. coli* cultures was purified using a Monarch miniprep kit (New England Biolabs, Ipswich, Massachusetts, USA) using the manufacturer's protocol. Isolated plasmids were quantified using a Nanodrop spectrophotometer, adjusted to a concentration of 500 ng/μL, and characterised by means of restriction digest —agarose gel electrophoresis assays (double-digestion with NotI-HF and BamHI-HF for pSFFV_sfCherry2(1–10), NotI-HF and HindIII-HF for pEGFP_TOMM20_sfCherry2(11); restriction enzymes from New England Biolabs).

Transient expression of sfCherry2$_{1-10}$ in NSCs was achieved as follows: $12 \times 10^6$ NSCs plated in a T75 flask (10-ml EV medium) were transfected with 250 μg of pSFFV_sfCherry2 (1–10) using Lipofectamine 3000 (Thermo Fisher Scientific) transfection reagent according to the supplier's protocol. After 48 hours incubation, the supernatant was collected and used for EV isolation, as described above, to obtain EVs$^{sfCherry2,1-10}$.

Transient expression of TOMM20-sfCherry2$_{11}$ in Mφ was achieved as follows. Bone marrow was flushed from femurs and tibiae, and bone marrow progenitor cells were cultured for 4 days on Petri dishes, as described above. At day 4, $1 \times 10^6$ bone marrow progenitor cells in the well of a 6-well plate (2-ml Mφ medium) were transfected with 5 μg of pEGFP_TOMM20_sf-Cherry2(11) using Lipofectamine LTX (Thermo Fisher Scientific) transfection reagent according to the supplier's protocol to obtain Mφ$^{sfCherry2,11}$. As a control, an additional $1 \times 10^6$ bone marrow progenitor cells were treated with Lipofectamine only (Mφ$^{LIPO\_CTRL}$). Differentiation of bone marrow progenitor cells was then continued for 2 additional days (6 days in total), and Mφ$^{sfCherry2,11}$ and control Mφ were then replated onto glass coverslips in a 12-well plate (100,000 cells/well).

Mφ$^{sfCherry2,11}$ and control Mφ were let to rest overnight and stimulated with 50 ng/ml LPS (Enzo Life Sciences). After 1 hour of LPS stimulation, cells were treated with EVs$^{sfCherry2,1-10}$ for 6 hours, washed with PBS 3 times, and fixed with 4% PFA. Coverslips were stained with primary conjugated TOMM20-Alexa 647 antibody (Abcam, ab209606, 1:500) and an anti-F4/80 rat primary antibody (Bio-Rad, 1:100) followed by an anti-rat 488 secondary antibody. Coverslips were washed 3 times with PBS and counterstained with the nuclear marker DAPI (MilliporeSigma, 1:10,00 in PBS) for 5 minutes at RT. Coverslips were washed 1 time with PBS before being mounted on glass microscope slides, and images were taken with a 63× objective on a confocal microscope (Leica TCS SP5 Microscope). Quantification of sfCherry2 signal [38] was obtained by analysing stacked images of $n = 10$ randomly selected cells per condition via ImageJ.

## Correlative-light electron microscopy

TEM finder grids (Agar Scientific, Stansted. UK, Maxtaform H-7) were placed individually in the centre of 13-mm diameter discs of Aclar (Agar Scientific), and the assembly was sputter coated with 60-nm carbon (Quorum 150T E carbon coater). The TEM finder grids were removed, placed into 24-well culture plates, and Mφ replated (30,000 cells/well) onto the carbon-side of the coverslips, left to rest overnight, and then stimulated with 50 ng/ml LPS (Enzo Life Sciences). After 1 hour of LPS stimulation, EVs isolated from MitoDsRed$^+$ NSCs were resuspended in Mφ medium and added to the same well (ratios: 1 Mφ: EVs collected from 30 MitoDsRed$^+$ NSCs). After 6 hours, the media was removed, the wells washed 1 time with 0.9% saline, and then placed in 500 μl of fixative solution (2% glutaraldehyde/2% formaldehyde in 0.05 M sodium cacodylate buffer pH 7.4 containing 2 mM calcium chloride) for 10 minutes. Grids were washed 3 times with wash/storage buffer (0.05 M sodium cacodylate buffer pH 7.4) to remove excess fixative and counterstained with DAPI (1:10,000 in 0.9% saline) for 5 minutes at RT. Grids were washed 3 times with wash/storage buffer, and epifluorescence/brightfield images were acquired using a 5× objective and a Leica epifluorescence microscope to identify ROIs for confocal imaging. One ROI per grid was then selected and Z-stacks (0.5 μm) were acquired using a 40× objective on a confocal microscope (Leica TCS SP5 Microscope).

Following fluorescence microscopy, samples were washed 5 times with 0.05 M sodium cacodylate buffer pH 7.4 and then osmicated (1% osmium tetroxide, 1.5% potassium ferricyanide, 0.05 M sodium cacodylate buffer pH 7.4) for 3 days at 4˚C. After washing 5 times in deionised water (DIW), samples were treated with 0.1% (w/v) thiocarbohydrazide/DIW for 20

minutes at RT in the dark. After washing 5 times in DIW, samples were osmicated a second time for 1 hour at RT (2% osmium tetroxide/DIW). After washing 5 times in DIW, samples were block stained with uranyl acetate (2% uranyl acetate in 0.05 M maleate buffer pH 5.5) for 3 days at 4˚C. Samples were washed 5 times in DIW and then dehydrated in a graded series of ethanol (50%/70%/95%/100%/100% dry), 100% dry acetone, and 100% dry ACN, 3 times in each for at least 5 minutes. Samples were infiltrated with a 50/50 mixture of 100% dry ACN/ Quetol resin mix (without BDMA) overnight, followed by 3 days in 100% Quetol resin mix (without BDMA). Then, the sample was infiltrated for 5 days in 100% Quetol resin mix with BDMA, exchanging the resin each day. The Quetol resin mixture is 12 g Quetol 651, 15.7 g NSA, 5.7 g MNA, and 0.5 g BDMA (all from Taab). Coverslips were then inverted cell/carbon-side down on top of resin-filled beam capsule lids, and the resin was set in a curing oven at 60˚C for 48 hours.

The samples were allowed to cool to RT, the beam capsule plastic was trimmed off using a razor blade, and samples were then mounted on blank resin stubs. The Aclar coverslips were pulled off gently, leaving the carbon imprint of the finder grid behind. The ROIs identified by fluorescence microscopy were located in the ultramicrotome (Leica Ultracut E) and trimmed using a razor blade. Thin sections were cut to a thickness of 90 nm using a diamond knife, mounted on Melinex plastic coverslips, and allowed to air dry. The Melinex coverslips were mounted on aluminium scanning electron microscopy (SEM) stubs using conductive carbon tabs (Taab), and the edges were painted with conductive silver paint. Then, samples were sputter coated with 30-nm carbon using a Quorum Q150 T E carbon coater. Samples were imaged in a Verios 460 scanning electron microscope (FEI/Thermo Fisher Scientific) at 3 to 4 keV accelerating voltage and 0.2 nA probe current in backscatter mode using the concentric back-scatter detector (CBS) in field-free/immersion mode for low-/high-resolution imaging, respectively; stitched image maps were acquired using FEI MAPS software. High-resolution maps were acquired at a magnification of 12kx, a working distance of 3.5 to 4 mm, an image pixel setting of 1536 x 1024, and a dwell time of 4 μs with 4-line integrations using the default stitching profile and 10% image overlap.

## Quantification of mitochondrial network morphology

Mφ were plated on coverslips (Microscope Cover Glasses, Glaswarenfabrik Karl Hecht GmbH, Sondheim, Germany) at 50,000 cells/well and the next day stimulated with 50 ng/ml LPS (Enzo Life Sciences). After 1 hour of LPS stimulation, Mφ were treated with or without EVs extracted from MitoDsRed$^+$ NSCs for another 6 hours, washed with PBS 3 times, and fixed with 4% PFA. Cells were then permeabilised with 0.1% Triton X-100, blocked in 5% normal goat serum, and stained with an anti-TOMM20 antibody (Santa Cruz, 1:300). Representative images were acquired as described above for the integration analysis. The mitochondrial network analysis was performed by an assessor blinded to the treatment conditions scoring each cell as fused, intermediate, or fragmented in respect to their mitochondrial network (average of $N \geq 140$ cells per $N = 6$ biological replicates).

## Extracellular flux (XF) assays

Mφ were seeded 6 days after bone marrow isolation with fresh Mφ medium in 24-well XF24 or 96-well XF96 cell culture microplates ($1 \times 10^5$ cells/well or $4 \times 10^4$ cells/well, respectively). Approximately 18 hours from seeding, Mφ were stimulated by adding 50 ng/ml LPS (Enzo Life Sciences). After 1 hour of LPS stimulation, EVs, exosomes, or EVs$^{Mito-depl.}$ extracted from NSCs (as detailed above) were resuspended in Mφ medium and added in the same well (ratios: 1 Mφ: EVs collected from 30 NSCs). At 6 hours from treatment, Mφ medium was replaced

with XF medium [Seahorse salt solution (Agilent Technologies), 1% glutamine 200 mM, 1% pyruvate 100 mM, 1% FBS, D-glucose (225 mg/50ml final volume)] pH 7.35–7.45, and baseline OCR and ECAR were measured for 3 reads (Mix: 3 minutes, Wait: 2 minutes, Measure: 3 minutes). Mitochondrial stress protocol was performed using oligomycin (1 μM), FCCP (1 μM), rotenone, and antimycin (1 μM) following manufacturer's instructions (XF cell mito stress test kit, Seahorse Bioscience).

For experiments in which EVs were preexposed to the uncoupling agent FCCP, EVs were isolated as previously described. After the ultracentrifugation step at 100,000 x $g$ for 70 minutes at 4˚C, EVs were resuspended in 1 ml of Mφ medium with 1 μM FCCP for 1 hour at 37˚C (EVs$^{FCCP}$) or Mφ medium alone (EVs). Particles were then washed with PBS and ultracentrifuged at 100,000 x $g$ for 30 minutes at 4˚C, followed by resuspension in 200 μl of PBS and an additional wash with 3.2 ml of PBS. Tubes were then subjected to an additional centrifugation at 100,000 x $g$ for 30 minutes at 4˚C using an optima MAX ultra with TLA-110 fixed angle rotor. The supernatant was completely discarded and EVs$^{FCCP}$ (or EVs) were finally resuspended in Mφ medium at the desired concentration.

After the completion of each XF assay, cells were washed with PBS, and 25 μl of 1x RIPA buffer (with protease/phosphatase inhibitors) were added to each well. The total protein amount/well was estimated with a BCA Protein Assay Kit (Thermo Fisher Scientific) and used to normalise the OCR and ECAR values of the single well.

## Quantitative gene expression analysis (qPCR) and microarrays

Mφ were plated in 12-well plates (100,000 cells/well), left to rest overnight, and stimulated with 50 ng/ml LPS (Enzo Life Sciences). After 1 hour of LPS stimulation, Mφ were treated with either NSC EVs, exosomes or EVs$^{Mito-depl.}$ in Mφ medium (ratios: 1 Mφ: EVs collected from 30 NSCs). Experiments in which EVs were preexposed to the uncoupling agent FCCP were performed as reported above. Total RNA was extracted 6 hours after treatment for qPCR using a RNeasy kit (Qiagen) according to the manufacturer's instructions. Total RNA (400 ng) was retro-transcribed into complementary DNA (cDNA) using a cDNA reverse transcription kit (Thermo Fisher Scientific) according to the manufacturer's instructions. qPCR was performed using TaqMan Universal PCR Master Mix (Thermo Fisher Scientific) and TaqMan Gene Expression Assays for *Il1* (Mm00434228_m1, Life Technologies), *Nos2* (Mm00440502_m1, Life Technologies), and *Il6* (Mm00446190_m1, Life Technologies). *Actb* (ACTB, REF4351315 Life Technologies) was used as internal calibrator. Samples were tested in triplicate on a Quant StudioTM 7 Flex Real-Time PCR System (Applied Biosystems) and analysed using the $2^{-\Delta\Delta CT}$ method. Briefly, the threshold cycle (CT) method uses the formula $2^{-\Delta\Delta CT}$ to calculate the expression of target genes normalised to a calibrator. The CT indicates the cycle number at which the amount of amplified target reaches a fixed threshold. The CT values were normalised for endogenous reference [$\Delta CT$ = CT (target gene) − CT (*Actb*, endogenous reference)] and compared with a calibrator using the $\Delta\Delta CT$ formula [$\Delta\Delta CT = \Delta CT$ (sample) − $\Delta CT$ (calibrator)].

For microarrays, samples were prepared according to Affymetrix protocols (Affymetrix, Santa Clara, California, USA). RNA quality and quantity were ensured using the Bioanalyzer (Agilent Technologies) and NanoDrop (Thermo Fisher Scientific), respectively. For RNA labelling, 200 ng of total RNA was used in conjunction with the Affymetrix recommended protocol for the Clariom_S chips. The hybridisation cocktail containing the fragmented and labelled cDNAs was hybridised to the Affymetrix Mouse Clariom_S GeneChip. The chips were washed and stained by the Affymetrix Fluidics Station using the standard format and protocols as described by Affymetrix. The probe arrays were stained with streptavidin

phycoerythrin solution (Thermo Fisher Scientific) and enhanced by using an antibody solution containing 0.5 mg/ml of biotinylated anti-streptavidin (Vector Laboratories, Burlingame, California, USA). An Affymetrix Gene Chip Scanner 3000 was used to scan the probe arrays. Gene expression intensities were calculated using Affymetrix AGCC software. Downstream analysis was conducted in R/Bioconductor [83].

The annotation package for the Clariom_S chips was retrieved from Bioconductor. The CEL files were then loaded into R, RMA normalised with the oligo package and filtered to only retain probes annotated as "main" [84]. Differential expression testing was performed using limma, and the resulting $p$-values were corrected with the Benjamini–Hochberg method. KEGG pathway enrichment analyses were performed using the GAGE package with default parameters [45]. Pathways of interest were visualised with Pathview [46]. The microarray raw data were deposited in ArrayExpress with the accession number E-MTAB-8250.

## EAE induction, treatment, and behavioural studies

C57BL/6 female mice were immunised with MOG-induced EAE, as previously described [37].

Body weight and EAE clinical score (0 = healthy; 1 = limp tail; 2 = ataxia and/or paresis of hind limbs; 3 = paralysis of hind limbs and/or paresis of forelimbs; 4 = tetraparalysis; and 5 = moribund or death) were recorded daily.

After 11 to 19 dpi, mice developed the first clinical signs of disease (disease onset). At 3 days after disease onset (i.e., PD), mice with similar scores were randomly assigned to the different treatment groups. After randomisation, each mouse received a single ICV injection (AP −0.15, ML +1.0 left, DV −2.4) of 5 μl PBS containing either $1 \times 10^6$ fGFP$^+$/MitoDsRed$^+$ NSCs ($n = 5$) or 64 μg/mouse of fGFP$^+$/MitoDsRed$^+$ EVs (either kept as total EVs [$n = 5$] or processed as EVs$^{Mito-depl.}$ [$n = 4$] and EVs$^{CD63-depl.}$ fractions [$n = 5$]). EAE mice receiving an ICV injection of 5 μl PBS alone [$n = 8$], and nonimmunised mice [$n = 4$] injected ICV with $1 \times 10^6$ fGFP$^+$/MitoDsRed$^+$ NSCs in 5 μl PBS, were used as controls.

## Ex vivo tissue pathology

At 55 dpi, mice were deeply anesthetised with an intraperitoneal (i.p.) injection of ketamine (Boehringer Ingelheim, Ingelheim am Rhein, Germany, 10 mg/ml) and xylazine (Bayer, Leverkusen, Germany, 1.17 mg/ml) in sterile water and transcardially perfused with 1 ml 5M EDTA in 500 ml saline (0.9% NaCl) for 5 minutes, followed by a solution of 4% PFA-PBS for another 5 minutes.

Brains and spinal cords were isolated and post-fixed in 4% PFA-PBS at 4˚C overnight. Tissues were then washed in PBS and left for at least 48–72 hours in 30% sucrose in PBS at 4˚C for cryo-protection. Brains and spinal cords were then embedded in optimum cutting temperature (OCT) medium, frozen with liquid nitrogen, and cryo-sectioned (25-μm coronal section thickness for brains and 10-μm axial section thickness for spinal cords) using a cryostat (Leica, CM1850) with a microtome blade (Feather, A35). Sections were then stored at −0˚C until use.

For quantification of mitochondrial transfer events, sections were rinsed with PBS, and then blocked for 1 hour at RT in blocking buffer (0.1% Triton X-100 and 10% secondary antibody species serum in PBS). A Fab fragment affinity purified IgG anti-mouse was applied if anti-mouse antibodies were used (Jackson ImmunoResearch Laboratories, Ely, UK, 1:10). The following primary antibodies, diluted in blocking buffer, were used at 4˚C overnight: anti-GFP (Invitrogen, 1:250), anti-DsRed (RFP) (Abcam, 1:400), anti-GFAP (Abcam, 1:500), anti-NeuN (MilliporeSigma, 1:250), anti-Olig2 (MilliporeSigma, 1:500), anti-CD3 (Abcam, 1:250), or anti-F4/80 (Bio-Rad, 1:100). Sections were then washed in PBS with 0.1% Triton X-100 and incubated with the appropriate fluorescent secondary antibodies (Invitrogen, Alexa-fluor 405,

488, 555, 647, 1:1,000) for 1 hour at RT. After washing in PBS, nuclei were counterstained with DAPI (Invitrogen, 1:10,000) for 3 minutes and then mounted with Dako mounting kit (Agilent Technologies). Nonspecific staining was observed in control incubations in which the primary antibodies were omitted.

Quantification of mitochondria transfer events in EAE was obtained from analysing $n \geq 12$ randomised brain ROIs per $N = 4$ biological replicates using a confocal microscope (Leica TCS SP5 Microscope) with a 40× objective, and data are expressed as % ± SEM of fGFP$^-$/MitoDsRed$^+$ particles co-localising with GFAP$^+$ astrocytes, NeuN$^+$ neurons, Olig2$^+$ oligodendrocytes, CD3$^+$ T cells, or F4/80$^+$ mononuclear phagocytes (an average of 292.5 total fGFP$^-$/MitoDsRed$^+$ particles per cell type were quantified).

## Statistical analysis

Statistical analyses of all data were performed with GraphPad Prism 9 (version 9.0.0 (86) for macOS, GraphPad Software, San Diego, California, USA). Differences between 2 groups were analysed using an unpaired $t$ test (unless otherwise stated). Differences among >2 groups were analysed using 1-way ANOVA followed by a Tukey multiple comparison test (unless otherwise stated). EAE scores, timed seahorse experiments, and changes in mitochondrial network were compared using 2-way ANOVA followed by a Bonferroni multiple comparisons test. Values are given in the text and figures as mean values ± SEM (unless otherwise stated), and a $p$-value $< 0.05$ was accepted as significant in all analyses (unless otherwise stated).

## Ethics statement and data policy

Animal research has been regulated under the Animals (Scientific Procedures) Act 1986 Amendment Regulations 2012 following ethical review by the University of Cambridge Animal Welfare and Ethical Review Body (AWERB). Animal work was covered by the PPL 80/2457 (to SP). The numerical data used in all figures are included in S3 Data.

## Supporting information

**S1 Fig. EV and mitochondria protein analysis of CD63 EV fractions and media. (a)** Densitometry analysis of CD63 protein expression in EVs$^{CD63\_enrich.}$ over EVs$^{CD63\_depl.}$ (± SEM). $N = 2$ independent experiments. (Data available in S3 Data). **(b)** Protein expression analysis by WB of EVs$^{CD63\_enrich.}$, EVs$^{CD63\_depl.}$, NSCs, EVs, NSC media, and EV media. Mitochondrial complexes proteins (CV-ATPase, CII-SDHB, CIV-MTCO1, CIII-UQCRC2, and CI-NDUFB8), EV enriched proteins (CD63 and CD9), and negative (Golga2) markers are shown, as well as β-actin. The same amount of input material was loaded into each well. EV, extracellular vesicle; NSC, neural stem cell; WB, western blot.
(TIF)

**S2 Fig. Size distribution and proteomic analysis of NSC EVs. (a)** Representative particle size distribution analysis of EVs by NTA, showing bimodal distribution of particle size in the exosome size range (30–150 nm) and in the submicron region (20–1,000 nm). Data are mean values (± SD). (Data available in S3 Data). **(b)** Comparison of EVs particle diameter (nm) and particle concentration (particle/ml) measurements using TRPS qNANO and NTA analysis. Data are expressed as mean values of $N = 3$ independent experiments. EV, extracellular vesicle; NSC, neural stem cell; NTA, nanoparticle tracking analysis; TRPS, tunable resistive pulse sensing.
(TIF)

**S3 Fig. OXPHOS genes modulated by EV in LPS-activated mononuclear phagocytes.** Pathview diagram showing the significantly enriched OXPHOS KEGG pathway. The colour scale represents the $\log_2$ fold change of each gene in EV-treated vs. untreated $M\varphi^{LPS}$ as measured by microarray analysis. (Data available on ArrayExpress, identifier E-MTAB-8250). EV, extracellular vesicle; KEGG, Kyoto Encyclopedia of Genes and Genomes; LPS, lipopolysaccharide; OXPHOS, oxidative phosphorylation.
(TIF)

**S4 Fig. Metabolism and pro-inflammatory activation of $M\varphi^{LPS}$ is not modified by exosomes or EVsMito_depl. treatment. (a)** XF assay of the OCR during a mitochondrial stress protocol of $M\varphi^{LPS}$ at 6 hours from treatment with exosomes or $EVs^{Mito\_depl}$ vs. $M\varphi^{LPS}$. Unstimulated $M\varphi$ were used as controls. Data are mean values (± SEM). $^{**}p < 0.01$ vs. $M\varphi^{LPS}$. $N = 5$ technical replicates per condition. (Data available in S3 Data). **(b)** Expression levels (qRT-PCR) of pro-inflammatory genes (*Il1β*, *Nos2*, and *Il6*) in $M\varphi^{LPS}$ at 6 hours from treatment with exosomes or $EVs^{Mito\_depl}$. Data are mean FI over $M\varphi^{LPS}$ (± SEM). $N = 2$ biological replicates from $N = 2$ independent experiments. $^{\#}p < 0.05$ vs. unstimulated $M\varphi$. (Data available in S3 Data). EV, extracellular vesicle; FI, fold induction; LPS, lipopolysaccharide; OCR, oxygen consumption rate; qRT-PCR, quantitative real-time polymerase chain reaction; XF, extracellular flux.
(TIF)

**S1 Video. Representative video of EV-associated mitochondria integrating in the host mitochondrial network in vitro.** EV-associated mitochondria (red, arrow) are shown to interact with the host mitochondrial network of $M\varphi^{LPS}$ (stained with MitoTracker Green FM). The x/y trace and 3D rendering show a representative MitoDsRed$^+$ mitochondria (red) fusing with host mitochondria (green). EV, extracellular vesicle; LPS, lipopolysaccharide.
(MP4)

**S1 Data. Complete dataset from the multiplex TMT-based proteomic experiments.** Complete dataset (unfiltered) from multiplex TMT-based proteomic experiment illustrated in Fig 1A. Normalised, unscaled protein abundances (biological replicates 1–3), log2(ratios), q values, and the numbers of unique peptides used for protein quantitation are shown in the "Complete proteomic dataset" worksheet. Data may be sorted and filtered by log2(ratios) or q values (columns F–K) to identify proteins significantly enriched in EVs (compared with NSCs) or exosomes (compared with NSCs or EVs). q values <0.05 are highlighted in red. In addition, data for any quantified protein of interest may be displayed in the "Gene search and plots" worksheet. Relative abundances (fraction of maximum, mean plus 95% CIs) for each condition are depicted by bars (grey, NSCs; gold, EVs; green, exosomes), as in Fig 1D. q values (highlighted in red if <0.05) and the number of unique peptides used for protein quantitation (with most confidence reserved for proteins with values >1) are also shown. CI, confidence interval; EV, whole EV fraction; Exo, sucrose gradient–purified exosomes; NSC, NSC whole-cell lysates; TMT, Tandem Mass Tag.
(XLSX)

**S2 Data. Complete dataset from the RNA microarray experiments.** Table showing the differential expression results from microarray analysis of $M\varphi^{LPS}$ treated with EVs vs. untreated $M\varphi^{LPS}$ at 6 hours. EV, extracellular vesicle; LPS, lipopolysaccharide.
(XLSX)

**S3 Data. Quantitative observations of data.** Excel spreadsheet containing, in separate sheets, the underlying numerical data and statistical analysis for Figs 1B, 1C, 1D, 3B, 3C, 4A, 4B, 4E, 4F,

4G, 5B, 5C, 6B, 6D, 6E, 6F, 7C, 8A, 8C, 8D, 8E, 9B and 9C and S1A, S2A, S4A and S4B Figs.
(XLSX)

## Acknowledgments

The authors wish to acknowledge G. Pluchino, G. Tannahill, B. Balzarotti, and J. Brelstaff for their technical contributions and critical insights throughout the execution of the study; C. Cossetti and R. Schulte of the CIMR Flow Cytometry Core Facility for their advice and support in flow cytometry; and D. Aubert and of A. Law of NanoFCM for their advice and support in nano flow cytometry analysis. The authors also thank A. Tolkovsky for providing the MitoDsRed plasmid and J. A. Enriquez (CNIC, Madrid Spain) for providing the L929-Rho$^0$ cells used in this study. We thank H. F. Greer for cryo-TEM data, the EPSRC Underpinning Multi-User Equipment Call (EP/P030467/1) for funding the TEM at the Department of Chemistry, and the Cambridge Advanced Imaging Centre (CAIC) for the CLEM processing and imaging.

## Author Contributions

**Conceptualization:** Luca Peruzzotti-Jametti, Joshua D. Bernstock, Christian Frezza, Stefano Pluchino.

**Data curation:** Luca Peruzzotti-Jametti, Cory M. Willis, Giulia Manferrari, Tommaso Leonardi, Ben Peacock, Edit Iren Buzas, Alain Brisson, Nicholas J. Matheson, Stefano Pluchino.

**Formal analysis:** Luca Peruzzotti-Jametti, Cory M. Willis, Giulia Manferrari, Erika Fernandez-Vizarra, James C. Williamson, Tommaso Leonardi, Ágnes Kittel, Carlos Bastos, Ben Peacock, Karin H. Muller, Nicholas J. Matheson, Carlo Viscomi, Stefano Pluchino.

**Funding acquisition:** Luca Peruzzotti-Jametti, Stefano Pluchino.

**Investigation:** Luca Peruzzotti-Jametti, Joshua D. Bernstock, Cory M. Willis, Giulia Manferrari, Rebecca Rogall, Erika Fernandez-Vizarra, James C. Williamson, Alice Braga, Aletta van den Bosch, Tommaso Leonardi, Grzegorz Krzak, Ágnes Kittel, Cristiane Benincá, Nunzio Vicario, Sisareuth Tan, Carlos Bastos, Iacopo Bicci, Karin H. Muller.

**Methodology:** Luca Peruzzotti-Jametti, Tommaso Leonardi, Nunzio Iraci, Jayden A. Smith, Ben Peacock, Edit Iren Buzas, Christian Frezza, Alain Brisson, Nicholas J. Matheson, Carlo Viscomi, Stefano Pluchino.

**Project administration:** Luca Peruzzotti-Jametti, Stefano Pluchino.

**Resources:** Luca Peruzzotti-Jametti, Paul J. Lehner, Massimo Zeviani, Carlo Viscomi, Stefano Pluchino.

**Supervision:** Luca Peruzzotti-Jametti, Nuno Faria, Alain Brisson, Carlo Viscomi, Stefano Pluchino.

**Visualization:** Luca Peruzzotti-Jametti.

**Writing – original draft:** Luca Peruzzotti-Jametti, Christian Frezza, Stefano Pluchino.

**Writing – review & editing:** Luca Peruzzotti-Jametti, Edit Iren Buzas, Stefano Pluchino.

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
