## [Editor Report · Decision Letter 0]

7 Feb 2020

Dear Dr Pluchino, 

Thank you for submitting your manuscript entitled "Neural stem cells traffic functional mitochondria via extracellular vesicles to correct mitochondrial dysfunction in target cells" for consideration as a Research Article by PLOS Biology.

Your manuscript has now been evaluated by the PLOS Biology editorial staff as well as by an academic editor with relevant expertise and I am writing to let you know that we would like to send your submission out for external peer review.

Please re-submit your manuscript within two working days, i.e. by Feb 09 2020 11:59PM.

Kind regards,

Di Jiang

PLOS Biology

---

## [Decision Letter · Decision Letter 1]

10 Mar 2020

Dear Dr Pluchino,

Thank you very much for submitting your manuscript "Neural stem cells traffic functional mitochondria via extracellular vesicles to correct mitochondrial dysfunction in target cells." for consideration as a Research Article at PLOS Biology. Your manuscript has been evaluated by the PLOS Biology editors, an Academic Editor with relevant expertise, and by three independent reviewers.

In light of the reviews (below), we will welcome re-submission of a much-revised version that addresses all reviewers' concerns. We cannot make any decision about publication until we have seen the revised manuscript and your response to the reviewers' comments. Your revised manuscript is also likely to be sent for further evaluation by the reviewers.

We expect to receive your revised manuscript within 4 months but if more time is need, we'd be happy to extend the deadline. 

**IMPORTANT - SUBMITTING YOUR REVISION**

*Re-submission Checklist*

*Published Peer Review*

*PLOS Data Policy*

*Blot and Gel Data Policy*

Sincerely,

Di Jiang, PhD

Associate Editor

PLOS Biology

REVIEWS:

Reviewer #1: In the manuscript by Peruzzotti-Jametti et al., the authors investigate intercell mitochondrial trafficking. The authors find that NSCs shed mitochondria/mitochondrial fragments via extracellular vesicles and these vesicles can be taken up by adjacent cells. The authors show that uptake of these EVs can improve mitochondrial function in Rho-zero cells as well as inflammatory macrophages. Finally, the authors show that in vivo transplant of NSCs, or injection of EVs can improve symptoms in a mouse model of MS.

This is a very intriguing manuscript that implies functional mitochondria can be transported between cells to influence their metabolism/behaviour. This has the potential to have significant implications for not just biology but also therapeutic approaches. However, I feel that the authors have not yet demonstrated that all the effects seen are due to mitochondrial transport and not some other element of EVs - and it may well take significant work to rule this out. Without solid data for mito-EVs, and not EVs in general, the manuscript too correlative.

Main points

1) The authors' data are clear in that NSCs can "shed" mitochondria, but what percentage of EVs actually contain mitochondria/mitochondrial fragments? The mito-EV preparations that the authors use also contain EVs that do not have mitochondria associated with them, so how are they certain that the effects seen are due to mitochondrial transfer itself? The authors need a control where they can generate the EVs without mitochondria and show these have no effects. I'm sure this is not a trivial matter and relates to the actual mechanism of how mitochondria get into these vesicles in the first place. The autophagy machinery has been linked to this - so perhaps inhibiting the initial stages of autophagy could produce EVs without mitochondria? Alternatively, could cells be depleted of mitochondria before collecting EVs (PMID: 28005069)?

2) The authors use dsRed-mito to localize mitochondria in recipient cells. Are the authors sure that this red signal is still mitochondrial? For example, the process of mitophagy could lead to delivery of dsRed tagged mitochondria to MVBs/late endosomes/lysosomes whereby partial digestion could occur to release the label from the mitochondria. If the MVBs/late endosomes/lysosomes then fuse with the plasma membrane, this could lead to release of dsRed unattached to mitochondria - which could then be taken up by other cells.

Related to this, in Figure 6 (and others) the authors show dsRed structures in the cell at various locations - including at the mitochondria. The authors imply that these dsRed-mitochondria become "integrated in the mitochondrial network", but how do they know this? The authors demonstrate that endocytosis is needed to uptake mito-EVs, thus they will be surrounded by at least one limiting membrane (maybe more if the EV membrane is taken into account) - how then do the mitochondria escape these membranes to become integrated into the network? My feeling is that these red structures could be endosomal in nature. Indeed, endosomes and lysosomes make extensive contacts with mitochondria without ever directly joining up with them.

The best way that I can think of to address these two points would be to carry out CLEM, which would show directly that the red dots are indeed mitochondria and that they integrate into the mitochondrial network and are not simply endocytic cargo.

3) I notice that for some of the imaging data, only the Imaris renderings are shown (e.g. Fig 6d). While these are very nice images, setting up the masking for the 3D mapping can be somewhat subjective, hence I think it is important to also include the original micrographs too. Related, for Figure 8a it looks like the three of the mitochondrial images are normal, while the M-LPS one has been processed through Imaris - they should all be the same. 

Reviewer #2: Neural stem cells traffic functional mitochondria via extracellular vesicles to correct mitochondrial dysfunction in target cells.

PBIOLOGY-D-20-00273R1

Reviewer's comments

General comments

The authors provide a well written, exciting paper that adds to the understanding of the transfer mechanism and functional relevance of extracellular vesicles/ectosomes-transmitted mitochondria from neural stem cells. The authors have presented a significant amount of data to unpick the role of EVs in shuttling mitochondria, covering the functionality in vitro and in vivo. Moreover, they have addressed a range of aspects of this phenomena, from the molecular functions of the mitochondria to identifying the recipient cells.

Major points

The authors have addressed some of the limitations of their techniques and methods in their discussion in detail. My major concern with this paper, however, has not been addressed, and that is the contribution of EV's present in the cell culture media. The NSC's are cultured in Stem Cell Technologies Neural basal media, with additional standard Proliferation Supplements. This contains human plasma, which in itself will be a source of EV's. Later experiments use mitochondrial Mito-DsRed fluorescent reporter to label NSC derived mitochondria, providing clear evidence that mitochondria are transferred from the NSC to a recipient cell. However, there is still the possibility of contaminating plasma-derived EVs in the in vivo experiments, for example, that could be contributing to the disease ameliorating effects seen. At the very least, the authors could use various analysis techniques to investigate the EV content of their culture media without NSC, and then confirm by Western blot that the profile of EV harvested from NSC grown in normal and EV-depleted culture media is the same. I appreciate that the NSC may not differentiate or grow well in EV-depleted media and that to repeat everything in EV-depleted media is not feasible. Still, it is essential to quantify the amount and effect of media-contributed EVs in this system. It would only require that the NSCs are grown in EV-depleted media for long enough to harvest NSC derived EVs. 

Furthermore, it is unclear to me how mitochondria-containing EVs are taken up in the receiving cells. Do the authors believe that the EV membrane fuses with the plasma membrane and the mitochondria are released in the cytoplasm? Are EVs endocytosed and routed to early endosomes? It is essnetial to address this point as it is important for the understanding of mitochondria function in the recepient cells. For example, the observations from the cryo-EM analysis could allow the authors to conclude how EVs/mitochondria are taken up.

To conclude, I would recommend the paper for publication after appropriate revision.

Minor points

- Based on a recent review from Raghu Kalluri and Valerie S. LeBleu in Sciences, 2020, microvesicles and apoptotic bodies belong to the so-called ectosomes. The authors may use the revised definition of extracellular vesicles in their work.

- Check the formatting of references throughout; numbering is not sequential. 

- In Fig 2a, some proteins such as CD9, beta-actin, etc., from the different EV preparation methods show different sizes. The authors need to clarify why this is. 

- Check the formatting of Fig. 2. The citations at the top are cut off. 

- At the bottom of page 22, relating to Fig 7, the authors should discuss the finding that cytochalasin D treatment also reduces the uptake of MitoDsRed EVs, not just D/P. The effect of cytochalasin D is, in fact, greater than D/P, but is not mentioned. 

Reviewer #3 (Eva-Maria Krämer-Albers, signed review): The manuscript by Peruzzotti-Jametti et al reports the observation that NSCs ship via Extracellular Vesicles (EVs) intact mitochondria (mito), which integrate into the mitochondrial network of LPS-stimulated macrophages, restoring damaged respiration and mitigating the proinflammatory phenotype. The data suggest that EV-mito transfer from NSCs to macrophages may play an immunomodulatory role under neuroinflammatory conditions and has the potential ability to ameliorate disease conditions such as those observed in EAE, the main mouse model of multiple sclerosis.

In my view, the study reveals novel aspects regarding the immunomodulatory functions of NSCs, which are of potential therapeutic relevance. The study comprises detailed molecular and morphological EV-characterization, in vitro functional analyses as well as promising in vivo data. However, while the transfer of functional mitochondria at least in vitro is convincingly demonstrated, I'm not entirely convinced that mitos indeed take their route via EVs (could be shuttled via another transcellular route). The study shows that EV-fractions contain free mitos and the presented experiments do not allow to discriminate the functional impact of free and EV-associated mitos. Thus, the route of delivery is not completely solved and either should be further addressed or better discussed.

Major points

1. Neither biochemical, morphological or functional data provide a final proof of EV-mediated transfer (see specific points below). This is partly due to technical limitations difficult to address. However, all functional experiments were performed with "crude" EV-fractions also containing free mitos. In the EAE experiment, multiple factors contained in the highly complex mito-EV-fraction may contribute to the observed amelioration. To provide further evidence of EV-mediated transfer a method involving EV-depletion should be employed (immune-mediated depletion or pharmacological/genetic manipulation of NSC to interfere with EV-release).

2. Alternative EV isolation protocols are used to support the data obtained with EVs isolated by differential centrifugation (Fig. 2). However, it should be clear that these are precipitation-based kits, which are known to co-isolate many other components (please add more information about the commercial EV isolation procedures at least to methods section). With regard to EV purity, a better alternative would be immuno-bead isolation (such as CD63 beads, which are commercially available). The Wang et al., 2017 method includes a 0.22 µm filtration step that should actually remove intact mitochondria, which typically range between 0.5 and 1 µm. The signals associated with this fraction thus likely reflect mitochondrial fragments contaminating the isolated EVs/exosomes.

3. Sucrose density gradient isolation was used to enrich for exosomes as a specific subtype of EVs and is assumed to provide EVs of higher purity due to the removal of non-EV components in the density gradient. Which criteria served to qualify fractions 6-9 as exosome containing fractions (please specify)? Indeed, Tsg101 WB shown in Fig. 3c indicates that fractions 5+6 rather qualify as exosome fractions, while fractions 7+8 may reflect high density non-exosomal material co-isolating with exosomes during differential centrifugation (Jeppesen et al, 2019). The mtDNA (Fig. 3d) spreads across the whole gradient (similar to actin) and it is difficult to see any enrichment in specific fractions rather looking like a non-specific association with all fractions. DNAse digestion could solve that question: In case of EV-association (or presence of intact free mitochondria) the DNA should be DNAse-resistant.

4. When screening for typical exosomal markers in the proteomics results (Table S1), these are enriched in the crude EV fraction, but rather depleted in what is considered as exosome fraction. On the other hand, the proteins enriched in the exosome fraction (compared to crude EVs) are largely mitochondrial proteins. Page 20 (last paragraph) refers to proteins selectively depleted in exosomes versus EVs. What are these proteins and is this indeed an argument of exosome-enrichment in the collected fractions? The paper contains valuable proteomic information that could be better presented to the reader. It would be extremely helpful for the reader if the authors could add some sorted data including a list of proteins enriched in EVs/NSCs, Exo/NSCs, and EVs/Exo (please define selection criteria). Furthermore, Venn diagrams showing overlaps between NSC, EVs, and Exo fractions would be informative and complement the volcano plots.

5. Fig. 4: EM images are not convincing. In Fig. 4a, I cannot identify EV-encapsulated mitochondria marked by the arrowhead. Also, it does not fit with the mean diameter reported in the graph shown below and the text (page 22). Can it be replaced by a more representative image? Is the mitochondrion depicted in Fig. 4b encapsulated in an EV? In fact, there should be three membranes. Furthermore, TOMM20 is a translocase sitting in the outer membrane. Most gold particles actually indicate inner membrane or even somatic localization of the detected antigen.

6. It is unclear from the study and discussion, which type of EVs (exosomes or plasma membrane derived EVs) the authors suggest to be of relevance for their findings. If I understand correctly, the authors argue that exosome-fractions also carry functional mitochondria. Have the authors tested whether the exosome fraction carry similar activity as crude EV-fractions (mito-EVs) regarding uptake and restoring mitochondrial functions in target cells (macrophages)? This would be a relevant addition to the manuscript.

7. Fig. 9 c+d are not conclusive and difficult to interpret. How were MitoDsRed particles exactly determined? The labelling of the y-axis in 9c is not clear. Confocal Localization of particles often overlap with nuclei (GFAP and NeuN stain). The appearance of the fGFP in the images should be explained (do the transplanted NSCs form such long processes?). It would actually be interesting to compare control mice that did not receive immunization.

Minor points

1. When screening TabS1 for EV-and Exo-enriched proteins, a number of proteins are turning up that are rather associated with differentiated neural cells (e.g. synaptic proteins and Proteolipid protein 1). How does this come about? Are the NSCs partly differentiated?

2. Fig. 1c and Fig. 3b: the dashed line actually indicates q<0.1 not 0.05

3. Fig. 1f, WB: why do EV markers Pdcd6ip and Tsg101 appear prominently in Mito fraction?

4. Methods: please add details regarding Accumax.

5. Fig. 5b, figure legend is unclear: how was quantified? I understood the percentage of cells taking up mito-EVs is depicted. Then, what does n=9 cells per replicate mean?

6. Fig. 6b, please label X-Axis of flow cytometry blots.

7. For the assays shown in Fig. 6+7, it would be interesting to see Mito in comparison to Mito-EVs. If the authors have the data, I would ask to share them with the community.

8. Fig. 9c,d: please specify analysed/depicted CNS region in the legend.

9. Please specify nature of error bars in legends.

---

## [Decision Letter · Decision Letter 2]

12 Feb 2021

Dear Stefano,

Thank you for submitting your revised Research Article entitled "Neural stem cells traffic functional mitochondria via extracellular vesicles to correct mitochondrial dysfunction in target cells." for publication in PLOS Biology. I have now obtained advice from the original reviewers 1 and 3 and have discussed their comments with the Academic Editor, who also carefully assessed the way you responded to reviewer 2.

Based on the reviews, we will probably accept this manuscript for publication, provided you will clarify the lack of actin or a different loading control in Figure S1b. 

Please also make sure to address the data and other policy-related requests listed below my signature.

We expect to receive your revised manuscript within two weeks. 

*Published Peer Review History*

*Early Version*

Sincerely,

Gabriel Gasque, Ph.D.,

Senior Editor,

ggasque@plos.org,

PLOS Biology

ETHICS STATEMENT:

-- Please include within your manuscript the full name of the IACUC/ethics committee that reviewed and approved the animal care and use protocol/permit/project license. Please also include an approval number.

-- Please include the specific national or international regulations/guidelines to which your animal care and use protocol adhered. Please note that institutional or accreditation organization guidelines (such as AAALAC) do not meet this requirement.

DATA POLICY:

--Please provide the dataset identifier for http://proteomecentral.proteomexchange.org. Please update your Data Availability statement to include such identifier.

--In addition to the microarray and proteomic data, we need you to provide as the individual quantitative observations that underlie the data summarized in the figures and results of your paper. For an example see here: http://www.plosbiology.org/article/info%3Adoi%2F10.1371%2Fjournal.pbio.1001908#s5

These data can be made available in one of the following forms:

Regardless of the method selected, please ensure that you provide the individual numerical values that underlie the summary data displayed in the following figure panels: Figures 1bcd, 3bc, 4abefg, 5bc, 6bdef, 7c, 8acde, 9bc, S1a, S2a, and S4ab.

Please also ensure that each figure legend in your manuscript includes information on where the underlying data can be found and that your supplemental data file/s has/have a legend.

Reviewer remarks:

Reviewer #1: The authors have extensively addressed my concerns.

Reviewer #3: The authors comprehensively addressed all reviewer comments and added several new convincing experimental data. The manuscript has significantly improved and all my concerns are fully satisfied.

---

## [Editor Report · Decision Letter 3]

1 Mar 2021

Dear Stefano,

Thank you for submitting your revised Research Article entitled "Neural stem cells traffic functional mitochondria via extracellular vesicles." for publication in PLOS Biology. Before I can pass your manuscript to the production department to formally accept the paper, I need you to do one more thing.

As mentioned in my previous decision letter, please ensure that each figure legend in your manuscript includes information on where the underlying data can be found: ArrayExpress, ProteomeXchang, and/or S3 Data.

We expect to receive your revised manuscript within two weeks. 

*Published Peer Review History*

*Early Version*

Sincerely,

Gabriel Gasque, Ph.D.,

Senior Editor,

ggasque@plos.org,

PLOS Biology

---

## [Editor Report · Decision Letter 4]

2 Mar 2021

Dear Dr Pluchino,

On behalf of my colleagues and the Academic Editor, Kate Storey, I am pleased to say that we can in principle offer to publish your Research Article "Neural stem cells traffic functional mitochondria via extracellular vesicles." in PLOS Biology, provided you address any remaining formatting and reporting issues. These will be detailed in an email that will follow this letter and that you will usually receive within 2-3 business days, during which time no action is required from you. Please note that we will not be able to formally accept your manuscript and schedule it for publication until you have made the required changes.

PRESS

Thank you again for supporting Open Access publishing. We look forward to publishing your paper in PLOS Biology. 

Sincerely, 

Gabriel Gasque, Ph.D. 

Senior Editor 

PLOS Biology